# Mutation in the mitochondrial chaperone TRAP1 leads to autism with more severe symptoms in males

Małgorzata Rydzanicz[1,12], Bozena Kuzniewska [2,3,12], Marta Magnowska[2,3], Tomasz Wójtowicz [4], Aleksandra Stawikowska [2,3], Anna Hojka[5], Ewa Borsuk[6], Ksenia Meyza[7], Olga Gewartowska[8], Jakub Gruchota[9], Jacek Miłek[2,3], Patrycja Wardaszka [2], Izabela Chojnicka [10], Ludwika Kondrakiewicz [7], Dorota Dymkowska[11], Alicja Puścian [7], Ewelina Knapska[7], Andrzej Dziembowski [6,9✉], Rafał Płoski [1✉] & Magdalena Dziembowska [2,3✉]

## Abstract

**There is increasing evidence of mitochondrial dysfunction in autism spectrum disorders (ASD), but the causal relationships are unclear. In an ASD patient whose identical twin was unaffected, we identified a postzygotic mosaic mutation p.Q639\* in the *TRAP1* gene, which encodes a mitochondrial chaperone of the HSP90 family. Additional screening of 176 unrelated ASD probands revealed an identical *TRAP1* variant in a male patient who had inherited it from a healthy mother. Notably, newly generated knock-in *Trap1* p.Q641\* mice display ASD-related behavioral abnormalities that are more pronounced in males than in females. Accordingly, *Trap1* p.Q641\* mutation also resulted in sex-specific changes in synaptic plasticity, the number of presynaptic mitochondria, and mitochondrial respiration. Thus, the *TRAP1* p.Q639\* mutation is the first example of a monogenic ASD caused by impaired mitochondrial protein homeostasis.**

**Keywords** Trap1; Autism; Mitochondria; Mouse Model; Synapses
**Subject Categories** Genetics, Gene Therapy & Genetic Disease; Organelles

## Introduction

Autism spectrum disorders (ASD) are a heterogeneous group of early-onset neurodevelopmental disorders with an estimated prevalence of ~1% in the general population and an average male-to-female ratio of 4–5:1 (Werling, 2016). ASD has a complex multifactorial etiology with a strong genetic contribution (Tick et al, 2016). Thanks to the development of large-scale whole-exome studies, many genes associated with ASD have been identified (Devlin and Scherer, 2012; Luo et al, 2018). Consequently, thousands of gene variants and mutations related to autism have been reported, and it is challenging to identify those most relevant to the disease. De novo mutations (Gaugler et al, 2014; Vicari et al, 2019) and postzygotic mosaic variants (PZMVs) (Dou et al, 2017; Freed and Pevsner, 2016; Krupp et al, 2017; Lim et al, 2017) also contribute to ASD incidence.

Most known human mutations related to autism have been identified in genes encoding synaptic proteins such as neuroligins, neurexins, synapsin 1, Shank proteins, and neurotransmitter receptors (Fassio et al, 2011; Monteiro and Feng, 2017; Sudhof, 2008). For some of these genes, mouse models with particular mutations have been generated and showed ASD-like behaviors and synaptic dysfunctions (Greco et al, 2013; Lim et al, 2021; Peca et al, 2011; Sacai et al, 2020; Tabuchi et al, 2007; Yoo et al, 2014). Analysis of these animal models has supported the concept that the pathophysiology of ASD is related to impaired synaptic function. Synapses are the neuronal regions with the highest energy consumption, making mitochondria particularly essential for synaptic functions. Accordingly, there is growing observational evidence of mitochondrial disturbance in autism spectrum disorders (ASD), but the causality of mitochondrial dysfunction in ASD remains to be determined.

Monozygotic twins (MZT), which arise from a single zygote and are considered physically and genetically identical, have been a valuable tool in understanding the extent of the genetic component of many diseases, including autism, for decades. Notably, any genetic differences between MZTs are expected to be postzygotic; thus, disease-discordant MZTs represent an exceptional opportunity to study direct genotype–phenotype correlations. Here, in an ASD-discordant MZT, we identified a p.Q639\* nonsense mutation in the *TRAP1* gene, which encodes a chaperone involved in mitochondrial protein homeostasis. Further analysis of a

[1]Department of Medical Genetics, Medical University of Warsaw, Warsaw, Poland. [2]Department of Animal Physiology, Faculty of Biology, University of Warsaw, Warsaw, Poland. [3]Centre of New Technologies, University of Warsaw, Warsaw, Poland. [4]Laboratory of Cell Biophysics, Nencki Institute of Experimental Biology, Warsaw, Poland. [5]Bioinformatics Core Facility, International Institute of Molecular and Cell Biology, Warsaw, Poland. [6]Department of Embryology, Faculty of Biology, University of Warsaw, Warsaw, Poland. [7]Laboratory of Emotions Neurobiology, Nencki Institute of Experimental Biology, Warsaw, Poland. [8]Genome Engineering Facility, International Institute of Molecular and Cell Biology, Warsaw, Poland. [9]Laboratory of RNA Biology, International Institute of Molecular and Cell Biology, Warsaw, Poland. [10]Department of Health and Rehabilitation Psychology, Faculty of Psychology, University of Warsaw, Warsaw, Poland. [11]Laboratory of Cellular Metabolism, Nencki Institute of Experimental Biology, Warsaw, Poland. [12]These authors contributed equally: Małgorzata Rydzanicz, Bozena Kuzniewska. ✉E-mail: adziembowski@iimcb.gov.pl; rafal.ploski@wum.edu.pl; m.dziembowska@cent.uw.edu.pl

corresponding knock-in mouse model revealed that the *TRAP1* p.Q639* variant leads to behavioral abnormalities more strongly presented in males than in females, accompanied by altered neuronal plasticity and dendritic spine morphology typical for ASD.

# Results

## A novel *TRAP1* p.Q639* variant in ASD-diagnosed monozygotic twin brother

We performed whole-exome sequencing on three ASD-discordant male MZT pairs (Table 1) with DNA samples obtained from hair follicles. In an ASD-affected twin, but not in his phenotypically normally developed brother, two potentially causal postzygotic mutations were found: a nonsense mutation p.Q639* in *TRAP1* (hg19: chr16:g.003712013-G > A, NM_016292.2: c.1915C>T; VAF 8%) and *RUVBL1* p.F329L missense mutation (hg19: chr3:g.127816172-A > C, NM_003707.2: c.987 T > G, p.F329L; VAF 48%) (Fig. 1A). Those were also not observed in blood samples of twins' parents; Fig. EV1). Both TRAP1 and RUVBL1 variants have a frequency of 0 in the gnomAD database and our in-house database of over 10,000 Polish individuals who underwent exome sequencing.

The *TRAP1* p.Q639* was in silico predicted as pathogenic by meta scores predictors (including BayesDel addAF, BayesDel noAF, 8 points) and five individual predictors (including CADD phred, EIGEN, EIGEN PC, FATHMM-MKL, GERP RS), as variant of uncertain significance by three individual predictors (including DANN, LRT, MutationTaster), and as benign by FATHMM-XF. Moreover, the C nucleotide at position c.1915 is highly conserved (PhyloP100 9.434) and Q residue at position 639 is also located in an evolutionarily conserved region of HSP90 domain of TRAP1 protein (Fig. 1E). In contrast, the *RUVBL1* p.F329L was predicted as benign by both meta scores and individual predictors. Furthermore, the T nucleotide at position c.987 is not conserved (PhyloP100 0.883).

The *TRAP1* p.Q639* variant introduces a premature stop codon, however the Z-score suggests that the gene is tolerant for loss-of-function (LoF) (−0.526). Thus, to assess the frequency of *TRAP1* LoF variants in the general population, we analyzed the distribution of LoF changes in the gnomAD database (v3.1.2). 61 different heterozygous LoF variants are reported in 202 individuals (total number of individuals >150,000; ~0.135%), including 42 (68.8%) located in the C-terminal HSP90 of TRAP1 protein (in 162 individuals, 80.2%). No significant difference in sex distribution was observed among *TRAP1* LoF variant carriers (95 males vs 107 females and 80 males vs 82 females when only HSP90 domain was considered).

Of note, in contrast to hair follicle DNA, both brothers were mosaic for the mutations in their blood samples (both with the same VAF: 2% for *TRAP1* variant and 22% for *RUVBL1*, respectively) (Fig. EV1A). Therefore, even though the *TRAP* and *RUVBL1* variants emerged post twinning in the ASD-affected brother, they were detected in the phenotypically normal twin blood sample. This may reflect postulated blood chimerism in MZTs due to the transfer of hematopoietic stem cells between the twins in utero, which mask genetic differences between disease-discordant MZTs when mutational screening on blood-derived DNA (Erlich, 2011; Forsberg et al, 2017), underscoring the importance of using non-hematopoietic tissues (Rydzanicz et al, 2022; Rydzanicz et al, 2021). Any genetic differences between MZTs are considered as postzygotic events. Since *TRAP1* p.Q639* mutation (VAF 8%) was found only in ASD-affected twin (when ectoderm-derived DNA was tested), thus it is considered as post-twinning event. The timing of mutation occurrence determines the variant distribution across different tissues (tissue-specific mosaicism) and VAF (Youssoufian and Pyeritz, 2002). The role of postzygotic mutations in ASD is well-established (Alonso-Gonzalez et al, 2021; D'Gama, 2021; Dou et al, 2017; Freed and Pevsner, 2016; Krupp et al, 2017; Lim et al, 2017). For identification of causative mosaic mutations, sequencing of DNA from affected tissue is recommended. The tissue which may be affected by postzygotic mutations in ASD patient is a brain available only postmortem (D'Gama et al, 2015; Woodbury-Smith et al, 2022). If affected tissue is inaccessible, then developmentally closest tissue that is available could be considered. Thus, in ASD-discordant MZTs we performed WES analysis using hair follicles derived from ectoderm, the same germ layer which contributes to brain. It was suggested that many postzygotic mutations with low VAF (>5%) observed in multiple tissues are likely to be mosaics in brain tissue as well (Jamuar et al, 2014).

To investigate the potential causative role of *TRAP1* and *RUVBL1* genes in influencing the risk of ASD, the entire coding sequence and intron–exon boundaries of both genes were analyzed in 176 unrelated ASD patients from Polish population (Table 2). No rare, functional variants were found in *RUVBL1*, while in four ASD patients variants in *TRAP1* were identified, including three heterozygous missense variants and one identical heterozygous nonsense p.Q639* *TRAP1* variant (VAF 50%). Interestingly, all *TRAP1* variants identified in the replication cohort of ASD patients were located within C-terminal HSP90 domain of TRAP1 protein (Fig. 1D; Table 3).

In the case of the second male ASD patient which had a similar psychological evaluation as the first patient (Fig. 1C), the *TRAP1* p.Q639* variant was inherited from a heterozygous phenotypically normal mother (Figs. 1B and EV1B). This could suggest sex-specific variable expressivity for this particular mutation. However, we could not establish the p.Q639* carrier status of the patient's grandparents. Given the *TRAP1* truncation variant was found in more than one ASD patient, we focused on this gene.

## Knock-in *Trap1* p.Q641* mice do not display a gross phenotype

The identified *TRAP1* p.Q639* mutation is predicted to generate a premature translation termination codon at position 639 of a mitochondrial chaperone of the HSP90 family. In order to investigate the potential mitochondrial etiology and sex-specificity of disease, we decided to employ an animal model of the *TRAP1* mutation. We generated a knock-in mouse with the identical *Trap1* mutation as that identified in two autistic patients (p.Q641* in mice is an equivalent of p.Q639* in humans). The mutation was introduced into the mouse genome using CRISPR-Cas9 (Fig. 2A,B). Both homozygous $Trap1^{Q641*/Q641*}$ (MUT) and heterozygous $Trap1^{WT/Q641*}$ (HET) animals were viable, fertile, and exhibited no

**Table 1. Related to Fig. 1. Results of psychological evaluation of ASD-discordant MZTs.**

| Patient | MZT-6_1 | MZT-6_2 | MZT-22_1 | MZT-22_2 | MZT-29_1 | MZT-29_2 |
|---|---|---|---|---|---|---|
| Age at sample collection | 5 years old | | 7 years old | | 9 years old | |
| Diagnosis | Childhood autism | None | Asperger's syndrome | None | Autism Spectrum Disorder | None |
| ADOS Classification | ASD | Non-spectrum | Autism | Non-spectrum | Autism | Non-spectrum |
| ADOS Communication | 3 | 3 | 6 | 1 | 4 | 2 |
| ADOS Social Interaction | 8 | 0 | 12 | 2 | 12 | 1 |
| ADOS Restricted and Repetitive Behavior | 5 | 0 | 2 | 0 | 4 | 0 |
| ADI-R Classification | Autism | Non-spectrum | Autism | Non-spectrum | Autism | Non-spectrum |
| ADI-R Social Interaction | 18 | 1 | 28 | 1 | 19 | 3 |
| ADI-R Communication | 9 | 2 | 24 | 3 | 12 | 4 |
| ADI-R Restricted and Repetitive Behavior | 5 | 0 | 11 | 0 | 7 | 0 |
| ADI-R Onset | 5 | 2 | 4 | 2 | 4 | 3 |

*ADOS* Autism Diagnostic Observation Schedule, *ADI-R* Autism Diagnostic Interview—Revised.
Bolded is ASD-affected twin.
Last two columns in italic refer to Patient 1 and his unaffected twin brother.

morphological abnormalities, consistent with the phenotype of ASD patients and with a prior report that *Trap1* is not essential in mice (Lisanti et al, 2014). Moreover, the *Trap1* mutation did not affect overall brain morphology (Appendix Fig. S1). We next examined the expression of *Trap1* in mouse male and female brains. Both mRNA and protein levels of *Trap1* were significantly reduced in heterozygotes, and truncated Trap1 protein was not detected in mutant mice (Fig. 2C,D; Appendix Fig. S2).

To obtain insight into how *Trap1* p.Q641* influences the cellular transcriptome of neuronal tissues, we have performed sequencing of RNA samples isolated from hippocampi of WT, $Trap1WT^{/Q641*}$, and $Trap1^{Q641*/Q641*}$ male and female mice. Surprisingly, we saw very minor changes in transcriptome profiles when comparing samples with mutated *Trap1* (either hetero- or homozygous) to samples with *Trap1WT* (Fig. 2E,F; Dataset EV1) and setting commonly used threshold value for differentially expressed genes (adj. *P* value < 0.05, |log2(FC)| <1). Notably, *Trap1* was the most dysregulated gene. At the same time, when we compared male to female samples grouped by *Trap1* mutation status, expected genes related to sex (*Ddx3y*, *Uty*, *Eif2s3y*, and *Xist*) were differentially expressed.

In the previous studies, homozygous deletion of TRAP1 in mouse hepatocytes and embryonic fibroblasts (MEFs) was shown to reprogram the transcriptome related to cellular bioenergetics (Lisanti et al, 2014). Since we have not observed significant changes in transcriptome profiles of $Trap1^{Q641*/Q641*}$ mice, we decided to broaden the list of differentially expressed genes by applying milder thresholding conditions (*P* value < 0.05, |log2(FC)| <0.38) between $Trap1WT^{/Q641*}$ and *Trap1WT* grouping samples by sex. Only a slight upregulation of genes with mitochondrial localization under *Trap1* was observed (Fig. EV2A,B) however, GO enrichment analysis did not uncover any specific pathway dysregulated under *Trap1* mutation. Interestingly, although transcriptional changes of genes encoding proteins with mitochondrial localization showed that the profiles are very similar in both sexes, there was one small cluster differentiating (Fig. EV2C).

Finally, since inhibition of *Trap1* by a small molecule (TPP) was shown to induce the mitochondrial unfolded protein response (UPR) and mitochondrial pre-RNA processing defects in cancer cells (Munch and Harper, 2016), in addition to mRNA coded in the nuclear genome we carefully analyzed mitochondrial RNAs within our dataset. As for the cytoplasmic mRNA, no dysregulation or evidence of mitochondrial stress on the transcriptome was detected (Appendix Fig. S3, related to Fig. 2). *Trap1* p.Q641* did not affect mitochondrial pre-RNA processing or lead to the induction of transcription factors and kinases involved in the UPR (Appendix Fig. S3, related to Fig. 2).

Together, our results showed that the *Trap1* p.Q641* mutation has a mild effect on the transcriptome with possible upregulation of genes involved in mitochondrial functioning.

## Social interaction deficits in *TRAP1* p.Q641* mice

To determine whether the mouse model recapitulates phenotypes relevant to ASD, we tested the mice for deficits in social interaction, one of the three ASD diagnostic criteria that can be replicated in mice (Silverman et al, 2010). Both individuals carrying the *TRAP1* p.Q639* mutation and diagnosed with ASD showed abnormal social interactions in ADOS and ADIR tests. To assess the potential social impairment of the mice, we used the Eco-HAB, an automated system for studying animal behavior (Puscian et al, 2016) (Fig. 3A) as well as a classical three-chamber test to study social approach. In both tests, males and females were examined separately. In the Eco-HAB the locomotor activity of $Trap1^{WT}$, $Trap1^{WT/Q641*}$, and $Trap1^{Q641*/Q641*}$ mice in consecutive days was the same for all genotypes and was typical for mouse circadian rhythm (Fig. 3B,C). Analysis of in-cohort sociability, which measures how much time each pair of animals living in the same cage spends together, revealed deficits in social interaction in male homozygous and heterozygous mutant mice compared with WT (Fig. 3D). *Trap1* MUT and HET males tend to spend significantly less time with other familiar mice than WT males. No such deficits in sociability

**A** Family 1

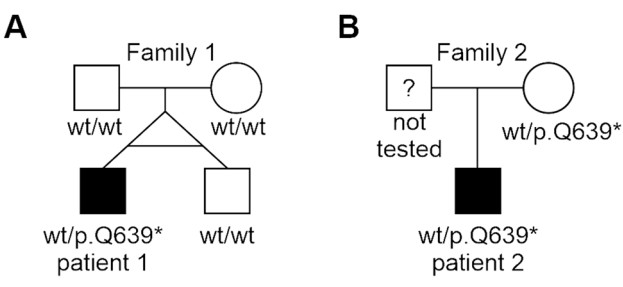

**B** Family 2

**C**

| Patient | | unaffected twin | patient 1 | patient 2 |
|---|---|---|---|---|
| Diagnosis | | None | ASD | ASD |
| **ADI-R** | Social Interaction | 3 | 19 | 28 |
| | Communication | 4 | 12 | 24 |
| | Restricted and Repetitive Behavior | 0 | 7 | 9 |
| | Onset | 2 | 4 | 5 |
| | Classification | Non-spectrum | Autism | Autism |

**D**

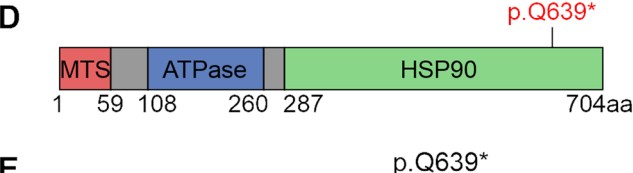

**E**

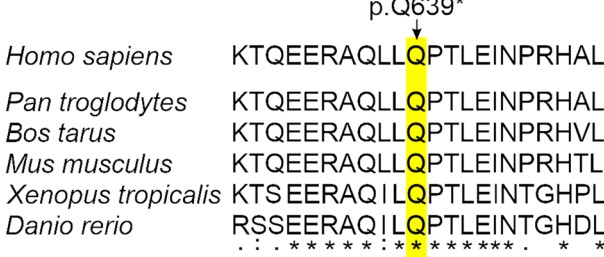

**Figure 1. Identification of a TRAP1 p.Q639* variant in ASD-affected individuals.**

Pedigree with phenotype and genotype denoted for (A) the examined MZT family and (B) the family of an additional ASD patient bearing the same TRAP1 mutation from the replication cohort; circle denotes female, square denotes male, symbol fill indicates ASD diagnosis. In a case of MZTs, genotype derives from hair follicle DNA samples. (C) Results of the psychological evaluation of the ASD- unaffected twin, patient 1 (ASD-discordant MZT) and patient 2 (from replication cohort) using the Autism Diagnostic Interview—Revisited (ADI-R). (D) Diagram depicting the location of identified p.Q639* variant in the TRAP1 protein. (E) Comparative sequence alignment of the TRAP1 protein in different species shows universal Q639 residue conservation in these species (highlighted).

**Table 2. Related to Fig. 1. Clinical characteristic of 176 unrelated ASD patients from the Polish population (replication cohort).**

| Characteristics | | | |
|---|---|---|---|
| Total number of patients | 176 | | |
| Gender | *n* (%) | | |
| Males | 149 (84.7%) | | |
| Females | 27 (15.3%) | | |
| Age of diagnosis | | | |
| Range | 12 months–9.5 years | | |
| Mean age ± SD | 3.92 ± 1.97 | | |
| **Phenotype classification**[a] | **Total** | **Males** | **Females** |
| | n (%) | n (%) | n (%) |
| F84.0 | 135 (76.7) | 109 (73.2) | 26 (96.3) |
| F84.1 | 2 (1.1) | 2 (1.3) | 0 |
| F84.5 | 16 (9.1) | 16 (10.7) | 0 |
| F84.8 | 1 (0.6) | 1 (0.7) | 0 |
| F84.9 | 22 (12.5) | 21 (14.1) | 1 (3.7) |

[a]Phenotype classification according to the International Statistical Classification of Diseases and Related Health Problems 10th Revision (ICD-10)—World Health Organization; Version:2016 (https://icd.who.int/browse10/2016/en). F84.0—Childhood autism, F84.1—Atypical autism, F84.5—Asperger syndrome, F84.8—other pervasive developmental disorders, F84.9—pervasive developmental disorder.

showed no preference for the unfamiliar mouse, unlike wild-type mice. The Trap1 MUT mice explored the non-social object and the new mouse equally (Fig. 3F–H).

The Eco-HAB-based sociability test examines social interactions among a group of familiar mice living together in one cage. In this test, social deficits were observed only in males, not in females. In contrast, the three-chamber social approach test, which evaluates interaction with an unfamiliar mouse, reveals disruptions in both male and female mutant mice. Overall, the behavioral phenotype is more pronounced in males than in females. These results are consistent with the diagnosis of ASD in one patient from the replication cohort but not in his mother, despite the presence of the *TRAP1* p.Q639* mutation in both, suggesting that the effect of this mutation is sex-related.

## Increased spine density and basal synaptic neurotransmission in *Trap1^{Q641*/ Q641*}* males

The synaptic imbalance observed in ASD is thought to give rise to functional changes in neural circuitry caused by altered properties of individual synapses (Zoghbi and Bear, 2012) located on dendritic spines undergoing plastic changes in response to synaptic stimulation. We analyzed spine morphology and electrophysiological recordings in the CA1 region of the hippocampus of WT, HET, and MUT male and female mice. For each mouse, one hemisphere was stained with DiI for assessment of the morphology of dendritic spines, while the second hemisphere was used for electrophysiological recordings. We observed striking differences between male and female brains in these assays. Male *Trap1^{Q641*/Q641*}* mice displayed increased spine density, but spines were shorter and had decreased area (Fig. 4A–E). In contrast, the effect of the mutation was the opposite in females, with slightly reduced spine density and

were observed in *Trap1* mutant females, indicating that the *Trap1* p.Q641* mutation in mice results in deficits in sociability specifically in males. In the three-chamber social approach test, designed to assess the preference for approaching a novel mouse over an inanimate object, both male and female Trap1 MUT mice

**Table 3.** Related to Fig. 1. Rare, functional *TRAP1* gene variants (frequency <0.0001) identified in replication and control groups.

| Variant (GRCh37/hg19) Effect (NM_016292.3) | ID | Number of individuals with SNV | gnomAD allele freq.[a] | Number of samples in DMG database[b] | Clinical symptoms, including ASD or DID | Pathogenicity prediction[c] | CADD phred score[d] | Comments |
|---|---|---|---|---|---|---|---|---|
| **Replication group (n = 176, unrelated ASD Polish patients)** | | | | | | | | |
| **chr16:003712013-G > A** | | **1** | **0** | **1** | **Yes** | **nd** | **46** | Patient 2, inherited from ASD-unaffected mother |
| **p.Q639*/c.1915C>T** | | | | | | | | |
| chr16:003712924-A > G | rs749530717 | 1 | 0.00005915 | 8 | No | T | 25.5 | Not present in mother, DNA from father not available for examination |
| p.W585R/c.1753 T > C | | | | | | | | |
| chr16:003714426-G > A | rs771557989 | 1 | 0.000006570 | 0 | - | T | 23.8 | Inherited from ASD-unaffected mother |
| p.S473L/c.1418 C > T | | | | | | | | |
| chr16:003714454-T > C | rs780375051 | 1 | 0 | 0 | - | T | 22.3 | Inherited from ASD-unaffected father |
| p.I464V /c.1390 A > G | | | | | | | | |
| **Control group (n = 100, unrelated healthy adult men from the Polish population)** | | | | | | | | |
| chr16:003708835-G > A | rs139636268 | 1 | 0.00002628 | 0 | - | T | 11.17 | |
| p.R658C/c.1972C>T | | | | | | | | |
| chr16:003727515-G > A | rs867558133 | 1 | 0.000006570 | 0 | - | nd | 12.93 | |
| p.Q230*/c.688 C > T | | | | | | | | |
| chr16:003736068-C > G | | 1 | 0 | 1 | - | T | 20.2 | Both SNVs observed in one individual (in cis) |
| p.D134H/c.400 G > C | | | | | | | | |
| chr16:003736071-A > G | | | 0 | | - | T | 19.77 | |
| p.S133P/c.397 T > C | | | | | | | | |
| chr16:003740966-G > A | rs140670575 | 1 | 0.000009200 | 0 | - | T | 8.896 | |
| p.R37W /c.109 C > T | | | | | | | | |
| chr16:003767494-G > A | | 1 | 0.00001317 | 0 | - | T | 10.75 | |
| p.R6W/c.16 C > T | | | | | | | | |

ASD autism spectrum disorder, *DID* disorders of intellectual development.
[a]Variant frequency according to gnomAD database v3.1.2 (http://gnomad.broadinstitute.org, accessed 2023-01-20); [b]in-house database of >10000 Polish individuals screened by WES (Department of Medical Genetics, Medical University of Warsaw); [c]missense variant's pathogenicity prediction based on MetaSVM, T (Tolerated); nd—no data, [d]Combined Annotation-Dependent Depletion (CADD) phred score. Bolded is variant identified in ASD-twin.

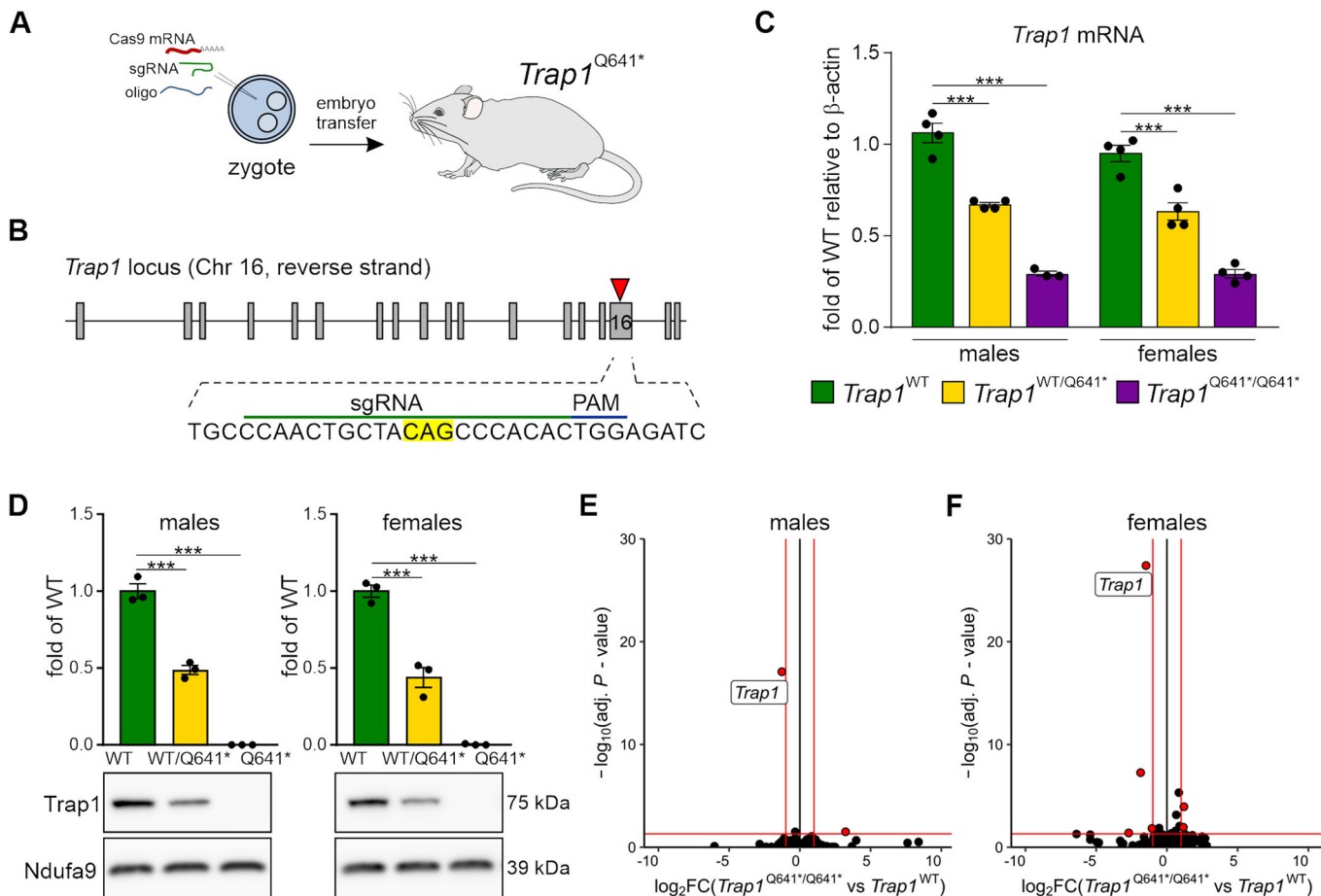

**Figure 2. Generation and transcriptomic analysis of a mouse with the proband *Trap1* mutation (p.Q639*) equivalent.**

(A) Schematic of the strategy to introduce a knock-in mutation into the *Trap1* locus using CRISPR/Cas9. (B) Diagram of sgRNA targeting site in the mouse *Trap1* locus. Codon for Q641 is highlighted; PAM protospacer adjacent motif. (C) *Trap1* mRNA levels in the cortex of *Trap1^WT^*, *Trap1^WT/Q641*^* and *Trap1^Q641*/Q641*^* mice as assessed by qPCR. Data are presented as fold change of *Trap1^WT^*, relative to *β-actin* mRNA; $n = 3$–4 animals/group; ***$P < 0.0001$, one-way ANOVA, post hoc Sidak's multiple comparisons test; error bars indicate SEM. (D) Trap1 protein levels in the hippocampus of *Trap1^WT^*, *Trap1^WT/Q641*^*, and *Trap1^Q641*/Q641*^* mice. Ndufa9 as loading control and SDS-PAGE TGX-gel are shown; $n = 3$ animals/group; ***$P < 0.0001$, one-way ANOVA, post hoc Sidak's multiple comparisons test; error bars indicate SEM. (E, F) Volcano plots representing the global differential gene expression in RNA-Seq analysis of the hippocampi of *Trap1^WT^* and *Trap1^Q641*/Q641*^* for male (E) and female (F) mice ($n = 3$–4 animals/group). The x-axis indicates log$_2$ fold changes (log$_2$FC) of gene expression levels in *Trap1^Q641*/Q641*^* versus *Trap1^WT^* mice, and the y-axis indicates –log$_{10}$ of adjusted $P$ value (adj.$P$ value). $P$ values were calculated with Wald test statistics and were adjusted with the Benjamini–Hochberg method. Black circles represent transcripts not differentially expressed, red circles represent transcripts significantly differentially expressed ($|log2(FC)| > 1$, adj.$P$ value $< 0.05$–thresholds designated by red lines on the plot). The topmost differential gene is labeled by gene symbol. See also Appendix Figs. S1 and S2. Source data are available online for this figure.

larger individual spines (Fig. 4F–J). The amygdala and medial prefrontal cortex (mPFC) are proposed to be one of several neural regions that are abnormal in autism (Baron-Cohen et al, 2000). Therefore, we analyzed dendritic spine density and morphology in mPFC and amygdala of WT, HET and MUT male and female mice, where we also saw a sex-specific difference. No alterations were observed in male mPFC, except for increased spine density in heterozygous mice (Appendix Fig. S4A–E). In contrast, females showed increased spine density, length, head width, and area (Appendix Fig. S4F–J). In the amygdala, close to no differences between analyzed groups were observed (Appendix Fig. S4K–T). Using the same mice, we investigated the extent to which the TRAP1 mutation affects basal excitatory synaptic transmission and short-term forms of synaptic plasticity in the CA1 hippocampal region of mice. We hypothesized that the increased density of dendritic spines in male

MUT mice could be reflected in changes of excitatory synaptic transmission. To test this possibility, we recorded AMPAR- and NMDAR-mediated field excitatory postsynaptic potentials (fEPSPs) in the CA1 hippocampal region (Fig. EV4A). AMPAR-mediated synaptic signals were significantly more pronounced in the hippocampus of male homozygous *Trap1^Q641*/Q641*^* mice compared with heterozygous and WT animals (Fig. 4K), consistent with the observed increased spine density. In contrast, female *Trap1^Q641*/Q641*^* mice with lower dendritic spine density showed decreased synaptic transmission (Fig. 4N). Male Trap1^Q641*/Q641*^ mice exhibited significantly larger NMDARs-mediated synaptic responses compared to WT and Trap1^WT/Q641*^ littermates (Fig. 4L). In contrast, female *Trap1^Q641*^* homo- and heterozygous mice exhibited reduced NMDAR-mediated synaptic transmission (Fig. 4O). The composition of postsynaptic glutamate receptors plays an essential role in the plasticity of the local

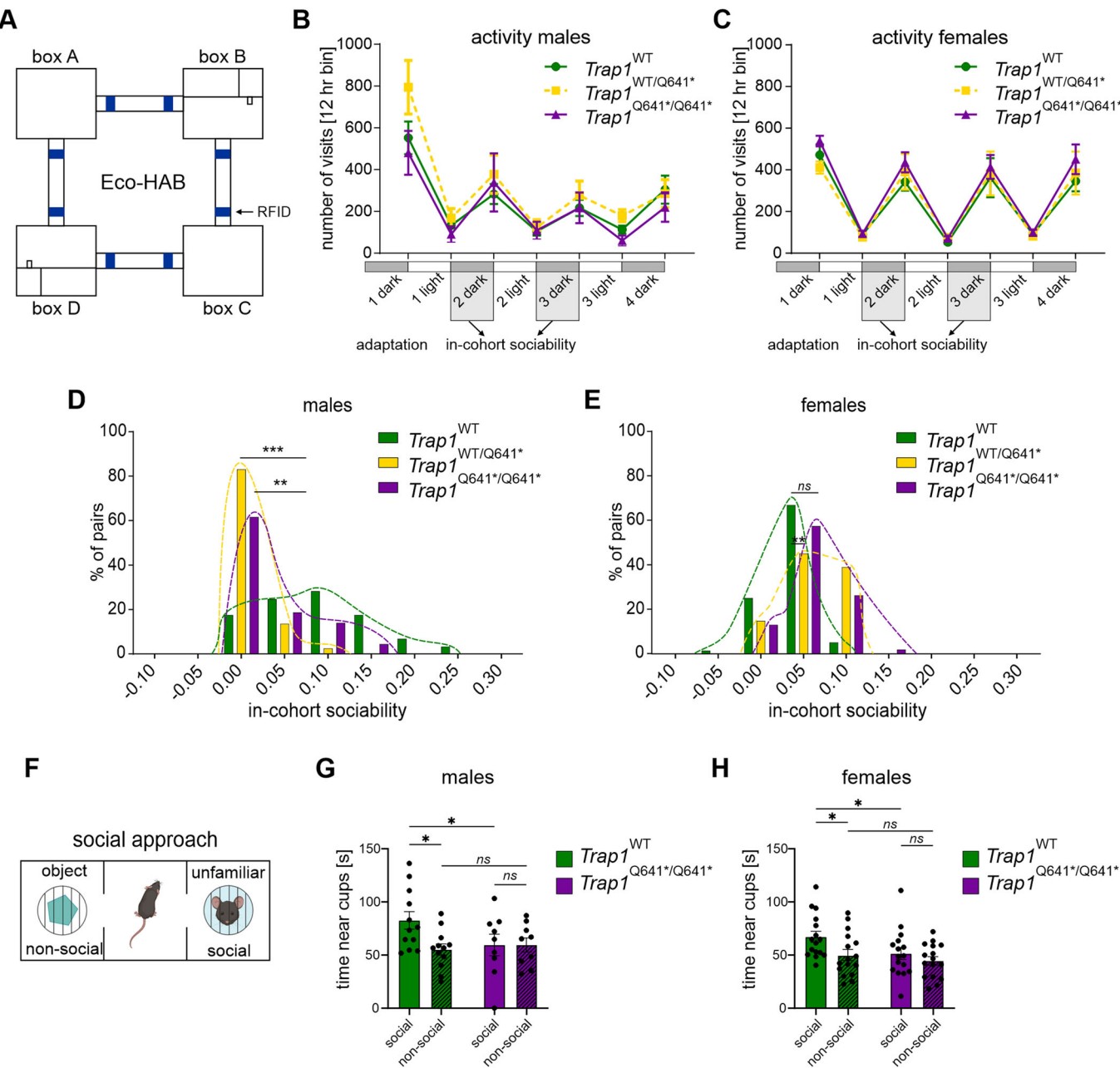

**Figure 3. Trap1 mutation in mice leads to more pronounced social deficits in males.**

(A) Schematic of the Eco-HAB apparatus, which consists of four housing compartments linked together with tube-shaped passages, where RFID antennas track mouse location. Food and water are available in two compartments: boxes (B, D). (B, C) Analysis of the locomotor activity of $Trap1^{WT}$, $Trap1^{WT/Q641*}$, and $Trap1^{Q641*/Q641*}$ mice of both sexes. No significant differences were observed. (D, E) Histograms show the distribution of "in-cohort sociability" parameters for all pairs of animals in the $Trap1^{WT}$, $Trap1^{WT/Q641*}$, and $Trap1^{Q641*/Q641*}$ cohorts in males (D) and females (E). In-cohort sociability measures how much time a pair of familiar animals voluntarily spends together. Data are presented as a relative frequency distribution histogram; $n = 7$–12 animals/group; ***$P < 0.001$, **$P < 0.01$; Kolmogorov–Smirnov test. (F) Scheme of the three-chamber social approach task. (G, H) Results of the social approach task of $Trap1^{WT}$ and $Trap1^{Q641*/Q641*}$ mice of both sexes. Both male and female Trap1WT mice show preference for the social cue (unfamiliar mouse) over non-social cue (*$P < 0.05$). In contrast, Trap1$^{Q641*/Q641*}$mice spend the same amount of time exploring social and non-social cues ($P > 0.05$) and less time exploring the social cue as compared to Trap1$^{WT}$ animals (*$P < 0.05$). $n = 9$–12 and 16–18 animals/group for males and females respectively; two-way ANOVA, post hoc uncorrected Fisher's LSD test; error bars indicate SEM).

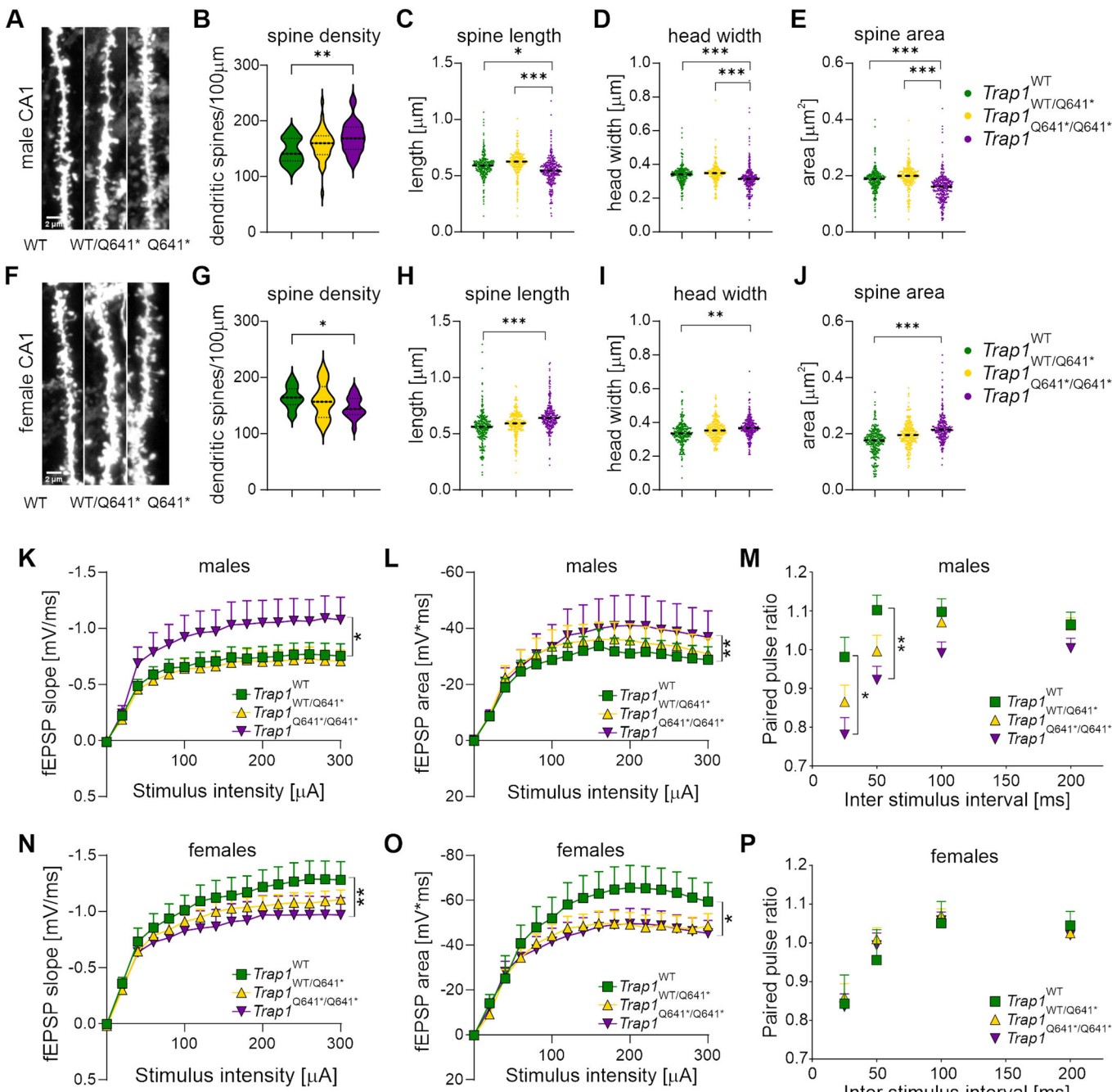

circuit. Altered availability of AMPARs and NMDARs in the synapse can lead to altered synaptic development, plasticity, and neuronal connectivity, resulting in synaptic pathophysiology (Uzunova et al, 2014). However, we found no changes in the sensitivity of fEPSPs to the AMPAR blocker DNQX in either sex of mouse regardless of genotype, suggesting that there is no gross change in the synaptic AMPAR/NMDAR ratio (Fig. EV4B,C).

Dysregulation of synaptic plasticity has been described in various animal models of autism (Filice et al, 2020; Lin et al, 2013; Takeuchi et al, 2013). To examine short-term synaptic

plasticity, we used paired-pulse facilitation, which results in scaling the synaptic responses proportional to the initial probability of presynaptic release (Regehr, 2012). We found that the average paired-pulse ratio for 25-ms and 50-ms interstimulus intervals was significantly reduced only in male $Trap1^{Q641*/Q641*}$ mice but not in female mice (Fig. 4M,P).

Together, our electrophysiological and neuroanatomical data show that male $Trap1^{Q641*/Q641*}$ mice specifically exhibit increased spine density and significant upregulation of basal synaptic neurotransmission in the hippocampus.

**Figure 4.  Altered dendritic spine density, morphology, and excitatory synaptic transmission in Trap1$^{Q641*/Q641*}$ mutant mice.**

(A) Representative images of DiI stained dendrites in the CA1 region of the hippocampus in males. Scale bars 2 μm. (B) The mean density of dendritic spines/100 μm of dendrite in the hippocampus in males. (C–E) Dendritic spine morphology in the CA1 region of the hippocampus, including spine length (C), spine head width (D), and spine area (E). (F) Representative images of DiI stained dendrites in the CA1 region of the hippocampus in females. Scale bars 2 μm. (G) The mean density of dendritic spines/100 μm of dendrite in the hippocampus in females. (H–J) Dendritic spine morphology in the CA1 region of the hippocampus, including spine length (H), spine head width (I), and spine area (J). (B, G) $n = 27$–38 (males), $n = 21$–31 (females) images/group; $*P < 0.05$, $**P < 0.001$; one-way ANOVA, post hoc Tukey's test (C–E, H–J) $n = 5274$–8555 (males), $n = 4010$–6118 (females) analyzed spines/experimental group; $*P < 0.05$, $**P < 0.001$, $***P < 0.0001$; nested ANOVA, post hoc Tukey's test. $N = 4$–6 animals/group (males), $N = 3$–4 animal/group (females). (K) Male Trap1$^{Q641*//Q641*}$ mice exhibited significantly enhanced AMPAR-mediated synaptic responses recorded in response to monotonically increasing stimuli compared to WT ($*P < 0.05$) and Trap1$^{WT/Q641*}$ ($***P < 0.001$) littermates (Kruskal–Wallis test, Dunn's Method for multiple pairwise comparisons). (L) Averaged synaptic responses recorded following monotonically increasing stimuli applied to presynaptic fibers. Male Trap1$^{Q641*/Q641*}$ mice exhibited significantly larger NMDARs-mediated synaptic responses compared to WT and Trap1$^{WT/Q641*}$ littermates (Kruskal–Wallis test, Dunn's Method for multiple pairwise comparisons, $**P < 0.001$). (M) The average paired-pulse ratio for 25 ms and 50 ms interstimulus intervals was significantly reduced in the male Trap1$^{Q641*/Q641*}$ group compared to both Trap1$^{WT/Q641*}$ and WT littermates (two-way RM ANOVA, Tukey's multiple comparisons test, $*P < 0.05$, $**P < 0.001$). (N) Female Trap1$^{WT/Q641*}$ and WT littermates exhibited significantly enhanced AMPAR-mediated synaptic transmission compared to the Trap1$^{Q641*/Q641*}$ group but were not different from each other (Kruskal–Wallis test, Dunn's Method for multiple pairwise comparisons, $**P < 0.001$). (O) Averaged synaptic responses recorded in females. Trap1$^{Q641*/Q641*}$ and Trap1$^{WT/Q641*}$ groups had significantly reduced NMDARs-mediated synaptic responses compared to wild-type littermates (Kruskal–Wallis test, Dunn's Method for multiple pairwise comparison, $*P < 0.05$). (P) Average paired-pulse ratios were not significantly different in any of the female genotype groups (two-way RM ANOVA, post hoc Tukey's test, $P > 0.05$). Males: $N = 3$–6 animals, $n = 12$–25 slices; Females: $N = 3$–4 animals, $n = 12$–17 slices. (K–P) Error bars indicate SEM. Source data are available online for this figure.

## Fewer presynaptic mitochondria and changes in mitochondrial metabolism in Trap1$^{Q641*/Q641*}$ mice

The Trap1$^{Q641*/Q641*}$ mutation exhibits male-specific phenotypes in mice, resulting in a reduced ratio of paired impulses, an electrophysiological phenomenon characteristic of impaired presynaptic release of neurotransmitters. Since the proper functions of the presynaptic compartment rely on stable and functional mitochondria (Rangaraju et al, 2019), we studied the CA1 region of the hippocampus in our mouse model using 3D electron microscopy. We examined the number, volume, and area of presynaptic mitochondria. Notably, we found that the density of presynaptic mitochondria was significantly lower in the CA1 hippocampus of male Trap1$^{Q641}$ mice, both homozygous and heterozygous (Fig. 5A–G). However, the average size and volume of individual mitochondria remained the same (Fig. 5H,I). These results, along with our previous finding of higher dendritic spine density in these male mice (Fig. 4B), suggest that some synapses may lack mitochondria in the presynaptic compartment.

To further explore mitochondrial metabolism in synapses, we used a simple model system (synaptoneurosomes) and measured electron flow rates in the electron transport chain (ETC) from 31 different mitochondrial substrates on mitoplates (Biolog) (Fig. EV5). Although there were no overall changes in mitochondrial metabolism when we compared males and females, we found significant differences in the use of the tricarboxylic acid cycle (TCA) substrates in males but not in females. Specifically, we observed decreased usage of succinate, the substrate for complex II and increased consumption of pyruvate + malate, substrates for complex I in Trap1$^{WT/Q641*}$ and Trap1$^{Q641*/Q641*}$ males compared to WT littermates (Fig. EV5C). No statistically significant differences were observed in females (Fig. EV5D). Both complex I and complex II pass electrons to complex III. In this context, the data obtained with MitoPlates suggested that potentially reduced activity of complex II may be balanced by the increased activity of complex I. Therefore, to verify this hypothesis we studied the respiration of mitochondria more precisely with high-resolution respirometry. Mitochondria were isolated from Trap1$^{WT}$ and Trap1$^{Q641*/Q641*}$ mice brains (cortex and hippocampus) and O$_2$ consumption rate was

measured O2k (Fig. 5J). Similarly to the Mitoplates, we found that in the presence of pyruvate, glutamate and malate, the mitochondria from Trap1$^{Q641*/Q641*}$ mice respired at a significantly higher rate than the Trap1$^{WT}$ (Fig. 5K,L). Moreover, male Trap1$^{Q641*/Q641*}$ mitochondria showed also a trend towards decreased respiration in the presence of ascorbate and TMPD, the substrates for complex IV when complex I and III were inhibited (Fig. 5L). Our results suggest that the activity of selected respiratory chain complexes is dysregulated in Trap1$^{Q641*/Q641*}$.

## Discussion

Mitochondria are crucial for the physiology of synapses since they provide ATP to meet their high energetic demands and are known to be essential for synaptic plasticity (Rangaraju et al, 2019). The link between ASD and mitochondria has been supported by clinical studies that reported disturbances at the levels of OXPHOS complexes activity, oxidative stress, and metabolites in the blood and urine of ASD patients (Hollis et al, 2017). Notably, ASD and mitochondrial disorders share common clinical features, and the prevalence of mitochondrial disease is 500 times higher in the ASD population (Cheng et al, 2017). At the same time, only a few genes encoding proteins related to mitochondrial function have been directly shown to be associated with ASD. These are neurofilament light polypeptide (NEFL), mitochondrial uncoupling protein 4 (Slc25A27), and mitochondrial aspartate/glutamate carrier gene (Slc25A12) (Anitha et al, 2012; Palmieri et al, 2010; Ramoz et al, 2004).

In this study, we identify and characterize a novel mutation in TRAP1 gene encoding HSP90 family mitochondrial chaperone. The HSP90 family members are ubiquitously expressed proteins, highly conserved among species. HSP90 are essential chaperones found in the cytoplasm and nucleus, maintaining cellular protein homeostasis. Trap1 is a mitochondrial matrix protein that shares 50% sequence similarity with other HSP90. However, the cellular function of Trap1 is still elusive, and there is very little data on its possible role in the nervous system. Trap1 was primarily studied in cancer cells, facilitating tumor progression through modulation

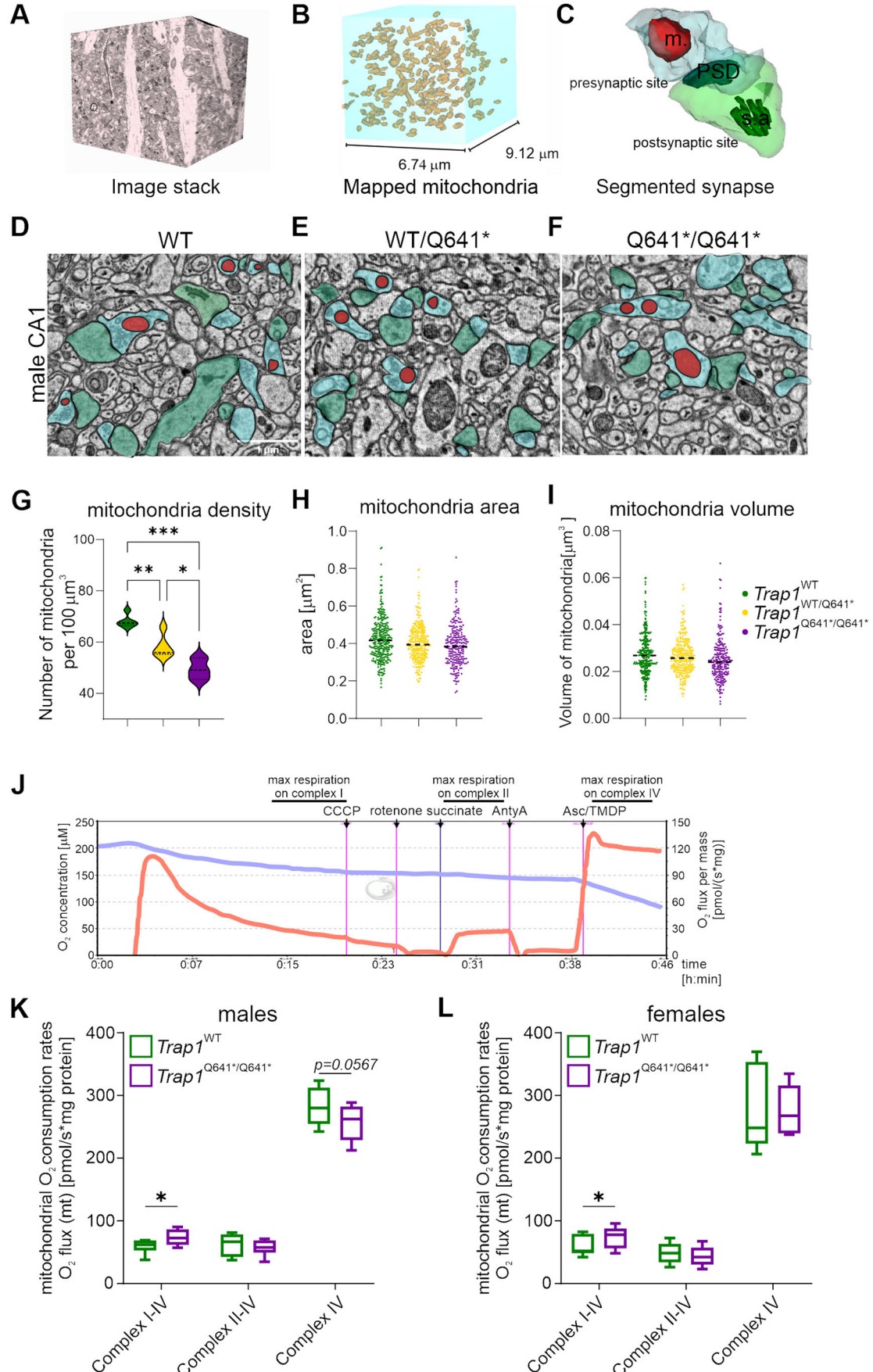

**Figure 5. Decreased number of presynaptic mitochondria in *Trap1^Q641*/Q641** and *Trap1^WT/Q641** male mice.**

(A) Representative Serial Block-Face Electron Microscopy (SBF-EM) image stack of stained striatum radiatum in the CA1 region of the hippocampus. (B) Illustration of mapped mitochondria in the image stack following manual segmentation. (C) An example of a segmented synapse with a presynaptic site (blue) with a mitochondrion (m, red), and a postsynaptic site (green) containing postsynaptic density (PSD) and spine apparatus (sa). (D–F) Representative electron micrographs of the CA1 region of the hippocampus in *Trap1* mice; pre- and postsynaptic sites with mitochondria are depicted as in (C). Scale bar 1 μm. (G) Decreased mitochondria density in *Trap1^Q641*/Q641** and *Trap1^WT/Q641** male mice. Plot shows mean density/100 μm$^3$; $N = 3$–5 animals/group; *$P < 0.05$, **$P < 0.01$, ***$P < 0.001$; one-way ANOVA, post hoc Tukey's. (H, I) No differences in mitochondria area (H) or volume (I) among tested genotypes. $P > 0.05$, nested ANOVA; $N = 3$–5 animals/group; $n = 200$–300 mitochondria/animal. (J–L) High-resolution respirometry of brain mitochondria (isolated from cortex and hippocampus) performed in an O2k Oxygraph. (J) Representative trace of oxygen concentration (blue line) and oxygen consumption (red line) as a function of time. Arrows indicate times of substrates and inhibitors addition. The following respiratory substrates were used: malate, pyruvate, and glutamate for complexes I–IV, succinate for complexes II–IV, and ascorbate and TMPD for complex IV. (K, L) Trap1^Q641*/Q641* male and female mitochondria show elevated levels of respiration in the presence of malate and pyruvate (*$P < 0.05$). Male Trap1^Q641*/Q641* mitochondria show also a trend towards decreased respiration in the presence of ascorbate and TMPD ($P = 0.0567$). Data are presented as a box-and-whiskers graph (the box extends from 25th to 75th percentiles, the central horizontal line is plotted at the median, and whiskers show 5th–95th percentile); $n = 6$ per genotype/sex, ratio paired two-tailed $t$ test. Source data are available online for this figure.

of cell metabolism (Chae et al, 2013; Purushottam Dharaskar et al, 2020; Yoshida et al, 2013). Studies on Trap1-deficient fibroblasts and hepatocytes showed that Trap1 is a regulator of mitochondrial bioenergetics and is implicated in metabolic reprogramming (Lisanti et al, 2014; Yoshida et al, 2013). Thus far, there is only one report linking *TRAP1* mutations with neurodevelopmental diseases, including ASD (Reuter et al, 2017). Reuter et al, reported *TRAP1* homozygous splicing variant in patient with moderate intellectual disability, mental deterioration, autism, self-mutilation, muscular hypotonia, nystagmus and leukodystrophy. Loss-of-function variants in *TRAP1*, including nonsense variants, were described in one late-onset Parkinson's disease patient (homozygous p.R47* (Fitzgerald et al, 2017), and in three patients with functional disorders (heterozygous p.E216*, p.Y229*, p.R703* (Boles et al, 2015). All previously reported nonsense variants are located within known functional domains: p.R47* is positioned within N-terminal mitochondria targeting sequence, p.E216* and p.Y229* within ATPase domain, and p.R703* within C-terminal HSP90 domain.

Here, we described a novel *Trap1* p.Q639* mutation, that is present in two unrelated male patients diagnosed with ASD, as well as in one unaffected mother. The identified mutation causes downregulation of the expression of Trap1 protein, leading to behavioral impairments that are more pronounced in males. The mechanism underlying the male bias in ASD is still not understood. However, several hypotheses are proposed to explain sex differences in the prevalence of the disorder i.e., female protective effect, male vulnerability, theories of sex hormones and a neuroendocrine hypothesis. Also, a multi-hit hypothesis, which includes genetic, environmental, and gender interactions, has also been proposed (review (Ferri et al, 2018)). As autism spectrum disorders are highly heterogeneous, these hypotheses do not have to exclude each other, and each hypothesis may account for certain ASD conditions.

The morphology of dendritic spines is determining their functionality and was shown to be changed in human and animal models of autism (Martinez-Cerdeno, 2017; Phillips and Pozzo-Miller, 2015). Our electrophysiological and neuroanatomical data show that male *Trap1^Q641*/Q641** mice specifically exhibit increased spine density and significant upregulation of basal synaptic neurotransmission in the hippocampus. Reduced paired-pulse ratios in these male MUT mice indicate decreased short-term synaptic facilitation. This may result from either increased basal neurotransmitter release probability (and thus reduced capacity for synaptic scaling) or impaired mechanism of

the presynaptic release. In practice, reduced paired-pulse facilitation may decrease the ability of *Trap1^Q641*/Q641** mice neurons to perform temporal summation of synaptic activity and propagate information following repetitive synaptic activity. An increase in the density of excitatory synapses, observed in *Trap1^Q641*/Q641** male mice could be a compensatory mechanism that ameliorates this disturbance in presynaptic sites. Moreover, we found that the density of presynaptic mitochondria was significantly lower in the CA1 hippocampus of male *Trap1^Q641** mice, suggesting that some synapses may lack mitochondria in the presynaptic compartment.

One hypothesis that could explain why the effects of TRAP1 mutation is more pronounced in males than in females is related to the potential sexual dimorphism of the synapse (Uhl et al, 2022). Studies suggest that in females the presence of estrogen receptors in synapses can modulate synaptic transmission, providing some protection (Huang and Woolley, 2012; Tabatadze et al, 2015). In addition, male neuronal mitochondria are more sensitive to oxidative stress, leading to respiratory dysfunction (Demarest et al, 2016). These factors may contribute to the heightened vulnerability and more pronounced behavioral phenotype in male Trap1 mutant mice.

Understanding the exact cause of the male bias in the *Trap1^Q641*/Q641** mice will require further investigation into the mitochondrial functions in neurons and with a specific focus on differences between genders. Our study highlights the necessity of in-depth analysis of both males and females in mouse models of ASD.

The etiology of ASD remains elusive, but recent studies, including the data presented, provide evidence for the role of mitochondrial homeostasis in autism. Thus, investigating strategies to modulate mitochondrial metabolism may hold promise for developing effective ASD treatments.

## Methods

### Ethical statement

Material was collected from the ASD-discordant MZTs (blood and hair follicles), ASD-affected patients (blood) and their parents (blood) after written informed parental consent was obtained. The study protocols were approved by the Institutional Review Board at Medical University of Warsaw (KB/153/2008 and KB/128/2014), the experiments conformed to the principles set out in the WMA

**Reagents and tools table**

| Reagent/resource | Reference or source | Identifier or catalog number |
| --- | --- | --- |
| **Experimental models** | | |
| B6.CBA/Tar (*M. musculus*) | This study | B6.CBA *Trap1*[em1limcb]/ Tar (*Trap1*[Q641*]) |
| **Antibodies** | | |
| Mouse anti-Trap1 | BD Biosciences | 612344 |
| Mouse anti-Ndufa9 | Abcam | ab14713 |
| Horse anti-mouse HRP | Cell Signaling Technology | 7076S |
| **Oligonucleotides and other sequence-based reagents** | | |
| sgRNA sequence: 5′-CCAACTGCTACAGCCCACAC-3′ | This study | |
| Trap1_Seq1F: 5′-TGTGTACCCTTGGACATCTT-3′ | This study | |
| Trap1_Seq1R: 5′-GAGAATGTCAGGTTTGTGCT-3′ | This study | |
| Trap1_mut_O1: 5′-ggagaccctgacaggtgtgtgctggagcccaggc ctacCTGGGGTTGATCTCCAGaGTtGGCTaTAGCAGTTGGGCAC GTTCCTCCTGGGTCTTGGCCAGCTGCTGCATACGCAAGAAAT-3′ | This study | |
| Trap1_sgRNA1F: 5′-gaaatTAATACGACTCACTATAGGGCCAACTGCTACAGCCCACAC GTTTTAGAGCTAGAAATAGCAAGTTAAAATAAGGC-3′ | This study | |
| Univ_IVsgRNA_2R 5′-AAAAGCACCGACTCGGTGCCACTTTTTCAAGTTGATAACG GACTAgccttattttaacttgctatttctagctcta-3′ | This study | |
| Trap1-1 Fw: TCCGCAGCATCTTCTATGTG | This study | |
| Trap1-1 Rev: TATACAGTGCCACGCTGGAG | This study | |
| Trap1-2 Fw: CAGGACAGTTATACAGCACACAG | This study | |
| Trap1-2 Rev: CTCATGTTTGGAGACAGAACCC | This study | |
| Actb Fw: CCCAGAGCAAGAGAGGTATC | This study | |
| Actb Rev: ATTGTAGAAGGTGTGGTGCC | This study | |
| **Chemicals, enzymes, and other reagents** | | |
| DNA IQ™ Casework Pro Kit for Maxwell® 16 | Promega | AS1240 |
| AmpFLSTR® NGM™ PCR Amplification Kit | Applied Biosystems | 4457889 |
| SureSelectQXT Reagent Kit | Agilent Technologies | |
| SureSelectXT Human All Exon V5 | Agilent Technologies | |
| Nextera XT DNA Library Preparation Kit | Illumina | |
| TruSeq Stranded Total RNA Library Prep | Illumina | 20020596 |
| adapters IDT for Illumina TruSeq RNA UD Indexes (96 Indexes, 96 Samples) | Illumina | 20022371 |
| 2100 Bioanalyzer and High Sensitivity DNA Kit | Agilent | 5067-4626 |
| KAPA Library Quantification Kit | Kapa Biosciences | KK4824 |
| NovaSeq 6000 S1 Reagent Kit | Illumina | 20012864 |
| protease inhibitor cocktail cOmplete EDTA-free | Roche | 5056489001 |
| RiboLock RNase Inhibitor | Thermo Fisher Scientific | EO0381 |
| 1 ml Dounce-type tissue grinder | WHEATON® | 357538 |
| Percoll | GE Healthcare | 17-5445-02 |
| PFA | Sigma-Aldrich | P6148 |
| Glutaraldehyde | Sigma-Aldrich | G5882 |
| Osmium tetroxide | Sigma-Aldrich | 75632 |
| Potassium ferrocyanide | Sigma-Aldrich | P3289 |
| Thiocarbohydrazide TCH | Sigma-Aldrich | 88535 |

| Reagent/resource | Reference or source | Identifier or catalog number |
|---|---|---|
| DMP-30 | Sigma-Aldrich | 45348 |
| Durcupan | Sigma-Aldrich | 44610 |
| Aclar sheets | Ted Pella | 10501-10 |
| Silver paint | Ted Pella | 16062-15 |
| Tungsten particles | Bio-Rad | |
| DiI | Invitrogen | 11520326 |
| MitoPlates S1 | Biolog | 14105 |
| TGX Stain-Free FastCast Acrylamide Solutions | Bio-Rad | 1610183, 1610185 |
| **Software** | | |
| GeneMapper ID v3.2.1 | Applied Biosystems | |
| bcl2fastq software | Illumina | |
| Burrows–Wheeler Alignment Tool | http://bio-bwa.sourceforge.net/ | |
| Picard | http://broadinstitute.github.io/picard/ | |
| Genome Analysis Toolkit | https://software.broadinstitute.org/gatk/ | |
| DatLab 7 | https://www.oroboros.at/index.php/product/datlab-7/ | |
| GraphPad Prism 9 | https://www.graphpad.com/ | |
| EthoVision XT 9 video-tracking system | Noldus, Wageningen, NL | |
| AxoGraphX | https://axograph.com/ | |
| pClamp10.7.0.3 | https://www.moleculardevices.com/products/axon-patch-clamp-system/acquisition-and-analysis-software/pclamp-software-suite | |
| ImageJ software | https://imagej.net/ | |
| **Other** | | |
| 31300xL Genetic Analyzer capillary sequencer | Applied Biosystems | |
| HiSeq1500 | Illumina | |
| MiSeq | Illumina | |
| Illumina NovaSeq 6000 sequencing platform | Illumina | |
| Eco-HAB system | Puscian et al (2016) | |
| Multiskan FC Microplate Photometer | Thermo Scientific | |
| Oxygrap-O2k | Oroboros Instruments | |
| SigmaVP | Zeiss | |
| Vibratome | Leica | VT1200S |
| Axio Imager Z2 LSM 700 | Zeiss | |
| Trans-Blot Turbo Blotting System | Bio-Rad | 170-4155 |
| Amersham Imager 600 | GE Healthcare | |

Declaration of Helsinki and the Department of Health and Human Services Belmont Report.

## Clinical evaluation of monozygotic twins (MZTs) discordant for ASD

Phenotypically abnormal twins have received a clinical diagnosis of ASD determined by a multidisciplinary team including a psychiatrist based on ICD-10 diagnostic criteria. Clinical evaluation was performed through parent interviews and MZT examination using Autism Diagnostic Interview—Revised (ADI-R) (Chojnicka and Pisula, 2019; Rutter et al, 2003) and Autism Diagnostic Observation Schedule (ADOS) (Lord et al, 2000) by a psychologist with experience in ASD diagnosis, as well as ADOS and ADI-R research reliability. All phenotypically abnormal twins met ADI-R and ADOS-2 criteria for an autism spectrum disorder.

## DNA extraction and zygosity testing

DNA from whole blood was purified using the standard salt-out method. DNA from hair follicles was extracted using the DNA IQ™ Casework Pro Kit for Maxwell® 16 (Promega, Madison, WI, USA). Twin zygosity was determined using blood genomic DNA and analyzed with AmpFLSTR® NGM™ PCR Amplification Kit (Applied Biosystems, Foster City, USA) to assess 17 highly polymorphic markers. PCR products were separated on 31300xL Genetic Analyzer capillary sequencer (Applied Biosystems, Foster City, CA, USA) and evaluated using GeneMapper ID v3.2.1 (Applied Biosystems). The obtained results were analyzed according to allelic ladder standards.

## Whole-exome sequencing (WES)

WES analysis was performed using 50 ng of genomic DNA extracted from hair follicles using the SureSelectQXT Reagent Kit and SureSelectXT Human All Exon V5 (Agilent Technologies, Cedar Creek, TX, USA) according to the manufacturer's instruction. Enriched libraries were paired-end sequenced ($2 \times 100$ bp) on HiSeq1500 (Illumina, San Diego, CA, USA). For all samples, >50 million read pairs were generated resulting in average mean depth >70×, coverage GE10 > 95% and GE20 > 90% of captured target.

WES sequencing data were analyzed as previously described (Rydzanicz et al, 2021). In brief, raw data was analyzed with bcl2fastq software (Illumina) to generate reads in fastq format. After the quality control step, including adapter trimming and low-quality reads removal, reads were aligned to the GRCh37 (hg19) reference genome with Burrows–Wheeler Alignment Tool (http://bio-bwa.sourceforge.net/) and processed further by Picard (http://broadinstitute.github.io/picard/) and Genome Analysis Toolkit (https://software.broadinstitute.org/gatk/). Base quality score recalibration, indel realignment and duplicate removal were executed, and SNV and INDEL discovery were performed. Identified variants were further annotated with functional information, frequency in population (including EXaC, gnomAD, dbSNP, dbNSFP, 1000 genomes, as well as the frequency from in-house database of >10,000 Polish individuals screened by WES), and known association with clinical phenotypes based on both ClinVar (Landrum et al, 2014) and HGMD (Stenson et al, 2003). In addition, for enhancing the sensitivity of low allele fraction (possible mosaic state) GATK MuTect2 was used (Cibulskis et al, 2013). Rare (frequency <0.01 in all tested databases) functional variants in protein-coding regions (missense, frameshift, stop-loss, stop-gain) and variants in splicing regions were considered. Priority was given to variants presented only in phenotypically abnormal twin (while in phenotypically normal co-twin lack of the variant was confirmed if covered >20×) or variants shared between twins but with significantly different variant allele fraction (effect of mutation load, incomplete penetrance). In silico pathogenicity prediction was performed based on Varsome (https://varsome.com) provided pathogenicity and conservation scores. All prioritized variants were manually inspected in Integrative Genomics Viewer (Robinson et al, 2011).

## Validation of single-nucleotide variants (SNVs)

Detected SNVs observed in the ASD-diagnosed twin but not in ASD-unaffected twin sibling were further validated by NGS-based deep amplicon sequencing (DAS). For validation analysis DNA samples isolated from blood (parents and MTZs) and hair follicles (MTZs) were tested. PCR primers for DAS were designed using Primer3 (http://primer3.ut.ee/) and are available upon request. In addition, for DAS to the locus-specific primers overhang adapters were added (forward overhang: 5'-TCGTCGGCAGCGTCA-GATGTGTATAAGAGACAG, reverse overhang: 5'-GTCTCGT-GGGCTCGGAGATGTGTATAAGAGACAG) to make the PCR product compatible with the Nextera XT index primers (Illumina). Amplicons-targeted identified SNVs were paired-end sequenced ($2 \times 100$ bp) on an Illumina HiSeq1500.

## Sequencing of *RUVBL1* and *TAP1* genes in a cohort of Polish ASD patients

For the replication study, the entire coding sequence and intron–exon boundaries of *RUVBL1* and *TRAP1* genes were analyzed using the NGS-DAS strategy in 176 unrelated ASD patients from the Polish population. Primers and PCR conditions are available upon request. For each patient, after PCR amplicon pooling, samples were subjected to NGS library preparation protocol using Nextera XT DNA Library Preparation Kit (Illumina), paired-end sequenced ($2 \times 250$ bp) on Illumina MiSeq and analyzed as described above for WES. Additionally, from the in-house WES database (>10,000 samples) 100 unrelated, healthy adult men from Polish population were selected as a control group, and rare SNVs distribution in *RUVBL1* and *TRAP1* genes were analyzed.

## Designing mice with point mutation in *Trap1* gene (*Trap1*(p.Q641*))

A new mouse line B6.CBA *Trap1*^em1Iimcb/Tar (*Trap1*^Q641*) was generated by the Mouse Genome Engineering Facility (www.crisprmice.eu). All mice were bred and maintained in the animal house of Faculty of Biology, University of Warsaw under a 12-h light/dark cycle with food and water available ad libitum. The animals were treated in accordance with the EU Directive 2010/63/EU for animal experiments.

Based on the mouse genome (GRCm38/mm10 Assembly) a single guide RNA (sgRNA) was designed using an online CRISPR tool (http://crispr.mit.edu). The chosen sequence did not show any major off-targets but had high calculated efficiency. For sgRNA synthesis, two oligodeoxynucleotides (ODN) carrying T7 polymerase promoter, guide sequence (TRAP1_sgRNA) and sgRNA scaffold (Univ_IVsgRNA_2R) were used to form dsDNA template for in vitro transcription using in-house T7 Polymerase. A 120 bp oligonucleotide Trap1_mut_O1 carrying point mutations to replace the wild-type Trap1 fragment was designed to introduce c.1921C>T mutation (p.Q641*) at the end of exon 16 of *Trap1*. Additionally two silent mutations: c.1926C>A (p.P642P) and c.1929A>T (p.T643T) were introduced, to minimize the risk of cleavage of recombinant DNA by Cas9. Cas9 mRNA was in vitro transcribed from Addgene pX458 plasmid using T7 RNA Polymerase. Poly(A) tail was added using E.coli PolyA Polymerase (Thermo Scientific) and m7Gppp5′N Cap was added with Vaccinia Capping System (NEB M2080) according to the manufacturer's protocols. Injection cocktail: (12.5 ng/µl Trap1_sgRNA IVT, 25 ng/µl Cas9 mRNA IVT, 1.5 pM Trap1_mut_O1) was introduced into mice zygotes via

microinjections. After 24–48 h of incubation, embryos were implanted into surrogate mice.

## Genotyping

Pups were genotyped at 4 weeks of age. DNA from tail or ear tips was isolated with Genomic Mini kit (A&A Biotechnology). gDNA was amplified with Trap1_Seq1F/1 R primer pair using Phusion HSII polymerase and HF buffer, then the amplicons were sequenced.

PCR cycling:
98′C—3:13
35× 98′C—13 s
63′C—17 s
72′C—7 s
72′C—5:00
12′C—forever

Primers used in the study:

sgRNA sequence: 5′CCAACTGCTACAGCCCACAC-3′

Trap1_Seq1F: 5′TGTGTACCCTTGGACATCTT-3′

Trap1_Seq1R: 5′GAGAATGTCAGGTTTGTGCT-3′

Trap1_mut_O1 (lowercase point mutations indicated): 5′-ggagaccctgacaggtgtgtgctggagcccaggcctacCTGGGGTTGATCTCCA-GaGTtGGCTaTAGCAGTTGGG-CACGTTCCTCCTGGGTCTTGGCCAGCTGCTGCATACGCAA-GAAAT-3′

Trap1_sgRNA1F (gRNA sequence in red):

5′-gaaatTAATACGACTCACTATAGGGCCAACTGCTA-CAGCCCACACGTTTTAGAGCTAGAAATAGCAAGTTAAAA-TAAGGC -3′

Univ_IVsgRNA_2R 5′-AAAAGCACCGACTCGGTGCCACTT-TTTCAAGTTGATAACGGACTAgccttatttaacttgctatttctagctcta-3′

## Animals

Mice were bred in the Animal House of the Faculty of Biology, University of Warsaw. The animals were kept in the laboratory animal facility under a 12-h light/dark cycle with food and water available ad libitum. The animals were treated in accordance with the EU Directive 2010/63/EU for animal experiments and Polish regulations. All experimental procedures were pre-approved by the Local Ethics Committee (WAW/194/2016 and WAW/771/2018 and WAW/1574P1/2024). The mouse lines can be provided by AD's pending scientific review and a completed material transfer agreement. Requests for the mouse lines should be submitted to AD.

## Tissue preparation

Young adult mice (~3 months old) were sacrificed, and the hippocampi and cortices were dissected and frozen at −80 °C for further investigation.

## SDS-PAGE and western blotting

Hippocampi were homogenized in RIPA buffer using a Dounce homogenizer. Protein content was measured using Pierce BCA protein assay kit. Equal amounts of samples were resolved by SDS-PAGE (10% or 12% TGX Stain-Free FastCast Acrylamide Solutions, Bio-Rad). After electrophoresis, proteins in the gel were visualized using Bio-Rad's ImageLab software to verify equal protein loading. Proteins were transferred to PVDF membranes (pore size 0.45 μm, Immobilon-P, Merck Millipore) using Trans-Blot Turbo Blotting System (Bio-Rad; 170-4155). Membranes were blocked for 1 h at room temperature in 5% non-fat dry milk in PBS-T (PBS with 0.01% Tween-20), followed by overnight incubation at 4 °C with primary antibodies in 5% milk in PBS-T (anti-Trap1, BD Biosciences Cat# 612344, RRID:AB_399710; anti-Ndufa9, Abcam Cat# ab14713, RRID:AB_301431). Blots were washed 3 × 5 min with PBS-T, incubated 1 h at room temperature with HRP-conjugated secondary antibody (1:10,000 in 5% milk), and washed 3 × 5 min with PBS-T. HRP signal was detected using Amersham ECL Prime Western Blotting Detection Reagent (GE Healthcare) on Amersham Imager 600 using automatic detection settings.

## RNA isolation and qRT-PCR

RNA was extracted from the mouse cortex using TRIzol (Thermo Fisher Scientific), DNA contamination was removed by 2 U of TURBO DNase (AM2238, Ambion) in the supplied buffer in 37 °C for 30 min. Next, RNA was re-isolated with phenol/chloroform, precipitated with ethanol, and resuspended in 50 μl of RNase-free water. RNA concentration was calculated from the absorbance at 260 nm using a DS-11 Spectrophotometer (DeNovix). Next, equal amounts of RNA samples (1.1 μg) were reverse transcribed using random primers (GeneON; #S300; 200 ng/RT reaction) and SuperScript IV Reverse Transcriptase (Thermo Fisher Scientific). Subsequently, the cDNA samples were amplified using sequence-specific primers in a final reaction volume of 15 μl, using PowerUp SybrGreen MasterMix in a LightCycler480 (Roche). Two different primer pairs were used to analyze *Trap1* mRNA: Trap1-1 Fw: TCCGCAGCATCTTCTATGTG; Trap1-1 Rev: TATACAGTGC-CACGCTGGAG; Trap1-2 Fw: CAGGACAGTTATACAGCACA-CAG; Trap1-2 Rev: CTCATGTTTGGAGACAGAACCC. Fold changes in expression were determined using the $\Delta\Delta$ Ct (where Ct is the threshold cycle) relative quantification method. Values were normalized to the relative amounts of *Actb* mRNA. Primer sequences: Actb Fw: CCCAGAGCAAGAGAGGTATC; Actb Rev: ATTGTAGAAGGTGTGGTGCC.

## RNA isolation, library preparation, and RNA sequencing

Total RNA was extracted from the mouse hippocampi with TRI Reagent (Sigma-Aldrich, Cat# 93289) according to the manufacturer's instructions and followed by DNase treatment (Invitrogen, Cat# 18080085). RNA quality and integrity was verified using RNA Pico 6000 (Agilent, Cat# 5067-1513). Strand-specific RNA libraries were prepared using a TruSeq Stranded Total RNA Library Prep (Illumina, Cat# 20020596) and adapters IDT for Illumina TruSeq RNA UD Indexes (96 Indexes, 96 Samples) (Illumina, Cat# 20022371) according to manufacturer's instructions. For library preparation, 1 μg of total RNA was used, fragmented by 8 min incubation at 94 °C. The library was enriched with 11 amplification cycles. The quality of the enriched library was verified using 2100 Bioanalyzer and High Sensitivity DNA Kit (Agilent, Cat# 5067-4626). The libraries' concentration was estimated by qPCR means with KAPA Library Quantification Kit (Kapa Biosciences, Cat# KK4824), according to the manufacturer's instructions. These

libraries were subsequently sequenced using an Illumina NovaSeq 6000 sequencing platform and NovaSeq 6000 S1 Reagent Kit (200 cycles) (Illumina, Cat# 20012864) in 2×100 nt pair-end mode with standard procedure according to the manufacturer's instructions.

## Quality control and mapping of high-throughput RNA sequencing data

The Illumina sequencing reads were quality filtered using Cutadapt v1.18 (https://doi.org/10.14806/ej.17.1.200 https://cutadapt.readthedocs.io/en/stable/) to remove Illumina adapter sequences, trim low-quality fragments (minimum Q score = 20), and remove reads that were shorter than 30 nt after trimming. After trimming, reads were repaired with bbmap v38.86 (https://doi.org/10.1371/journal.pone.0185056 BBMap— Bushnell B.—sourceforge.net/projects/bbmap/). Repaired reads were mapped to mouse genome (GRCm38) using STAR v2.7.5c (https://doi.org/10.1002/0471250953.bi1114s51 https://github.com/alexdobin/STAR). For downstream analysis, only uniquely mapped reads were used. The feature assignments of reads was performed with HTSeq v0.12.4 (https://doi.org/10.1093/bioinformatics/btac166 https://github.com/htseq/htseq), choosing genes as representative features. The normalization and differential gene expression analysis were performed with DESeq2 R package (https://doi.org/10.1186/s13059-014-0550-8). For mitochondrial junction analysis, a custom annotation of junctions was generated based on mitochondrial gene annotation from Gencode basic annotation v25. The count of junctions was generated as for gene features with HTSeq. For RNA-Seq analysis, all plots were generated in R/Bioconductor environment with the usage of ggplot2 package (Wickham H, 2016).

## Nissl-staining

Mouse brains (from 3-month-old mice) were fixed in 4% paraformaldehyde in PBS overnight at 4 °C, then the brains were cryoprotected in 20% sucrose in PBS at 4 °C for 48 h and frozen in −80 °C. Next, the brains were cut coronally on cryostat (Cryostat Leica CM 1860) on 40-μm slices. Coronal sections were air-dried on slides and stained with 0.1% cresyl violet solution (containing 3% acetic acid) for 5 min, washed, dehydrated, cleared in xylene, and coverslipped.

## Eco-HAB

Eco-HAB experiments were performed on young adult male and female mice (~2.5- to 4-month-old). Experiments were performed as previously described (Puscian et al, 2016). To individually identify animals in Eco-HAB, all mice were subcutaneously injected with glass-covered microtransponders (9.5 mm length, 2.2 mm diameter, RF*IP* Ltd) under brief isoflurane anesthesia. Microtransponders emit a unique animal identification code when in the range of RFID antennas. After the injection of transponders, subjects were moved from the housing facilities to the experimental rooms and adapted to the shifted light/dark cycle of their new environment (the dark phase shifted from 20:00–8:00 to 13:00–01:00 or 12:00–24:00 depending on summer/winter UTC + 01:00). For 2 weeks prior to behavioral testing, subjects were housed together and grouped appropriately for their respective experiment. The following cohorts were used: (a). males: WT, $n = 8$;

HET, $n = 9$; MUT, $n = 7$, (b) females: WT, $n = 11$; HET, $n = 12$; MUT, $n = 10$. Cohorts were subjected to 84-h Eco-HAB testing protocols (see Figs. 3A and EV4A) divided into an adaptation phase (24 h) and in-cohort sociability testing phase (next 48 h). Throughout the experiment, mice could freely explore all compartments, with unrestricted access to food and water in two of the housing compartments. All the details concerning Eco-HAB apparatus construction, software package for data collection, processing and analysis is described in the original paper (Puscian et al, 2016). Briefly, activity was defined as the number of visits of an animal to all of Eco-HAB compartments in 12 h bins. In-cohort sociability of each pair of mice within a given cohort is a measure of sociability that is unique to the Eco-HAB system.

For each pair of subjects within a cohort, i.e., animal a and animal b, the times spent by the mice in each of the four compartments during a chosen experimental period (48 h, 12 h bins) were calculated: $t_{a1}, t_{a2}, t_{a3}, t_{a4}$ for animal a, and $t_{b1}, t_{b2}, t_{b3}, t_{b4}$ for animal b. Next, the total time spent by the pair together in each of the cages was calculated: $t_{ab} = t_{ab1} + {;}t_{ab2} + t_{ab3} + t_{ab4}$. All times are normalized by the total time of the analyzed segment, so that each of the quantities fall between 0 and 1. The in-cohort sociability was then defined by $t_{ab} − (t_{a1}*t_{b1} + t_{a2}*t_{b2} + t_{a3}*t_{b3} + t_{a4}*t_{b4})$, which is the total time spent together minus the time animals would spend together assuming independent exploration of the apparatus (for details see the original paper (Puscian et al, 2016)).

## Three-chamber social approach test

3-chamber tests were performed on young adult male and female mice (~2.5- to 5-month-old). Experiments were performed as previously described (Puscian et al, 2016). Mice were habituated to the experimenter and handling procedures for 14 days prior to testing. The three-chambered apparatus (length—820 mm, width—420 mm, height—410 mm, center chamber size—200 mm, side chambers—300 mm each; door width—80 mm; 10 mm thick walls made of gray foamed PVC) was used. The center area was object-free, while the side areas contained either a social or a non-social stimulus placed in small steel cages (length—100 mm, width—100 mm, height—110 mm). The protocol for the assessment of social preference consisted of three sessions: (1) habituation to the center chamber, (2) habituation to all three empty chambers, and (3) a testing session when social (an unfamiliar mouse of the same strain and sex) and non-social stimuli (a novel green plastic bottle cap) were presented in side chambers. Each session lasted 10 min and was video-recorded. The video was analyzed using a EthoVision XT 9 video-tracking system (Noldus, Wageningen, NL) to extract behavioral data. The "near cup" zone was defined as $140 × 140$ mm constituting a 2 cm border zone around the edge of the cup. The following parameters were taken into analysis— distance moved in the center area with doors closed during habituation session, time spent in each of the compartment: center and empty side chambers (future social and future non-social) during habituation session with doors open and finally: time spent near cups in the social approach session. We excluded data from the analysis corresponding to the situation in which an animal had not visited both side chambers in either adaptation or social preference testing phase. In other words, we discarded those rare cases in which the animals' locomotor activity was extremely low.

## DiI staining of brain slices

To visualize changes in the shape of dendritic spines, 1,1′-dioctadecyl-3,3,3′,3′-tetramethylindocarbocyanine perchlorate (DiI) staining was performed in brain sections from *Trap1* mutant mice, heterozygotes and wild-type mice. The mice were anesthetized and transcardially perfused with 1.5% paraformaldehyde. The brains were dissected and sliced using a vibratome. Slices (100 μm thick) that contained the hippocampus were allowed to recover for at least 1 h at room temperature. Random dendrite labeling was performed using 1.6 μm tungsten particles (Bio-Rad, Hercules, CA, USA) that were coated with propelled lipophilic fluorescent dye (DiI; Invitrogen) that was delivered to the cells by gene gun (Bio-Rad) bombardment. Images of dendrites in hippocampal CA1, medial prefrontal cortex, and amygdala were acquired under 561 nm fluorescent illumination using a confocal microscope Axio Imager Z2 Zeiss LSM 700 (63× objective, 1.4 NA) at a pixel resolution of 1024 × 1024 with a 3.4 zoom, resulting in a 0.07 μm pixel size. Images were acquired blinded to the genotype group.

## Morphometric analysis of dendritic spines

The analysis of dendritic spine morphology and calculation of changes in spine shape parameters were performed as described previously (Magnowska et al, 2016; Michaluk et al, 2011). The images acquired from the brain slices were processed using ImageJ software (National Institutes of Health, Bethesda, MD, USA) and analyzed semiautomatically using custom-written SpineMagick software (patent no. WO/2013/021001) blinded to the genotype group. The analyzed dendritic spines belonged to secondary and ternary dendrites. We used length, head width, area and a scale-free parameter of relative changes in the spine length-to-head width ratio, which reflects spine shape. The spine length was determined by measuring the curvilinear length along a fitted virtual skeleton of the spine. The fitting procedure was performed by looking for a curve along which integrated fluorescence was at a maximum. The head width was defined as the diameter of the largest spine section while excluding the bottom part of the spine (1/3 of the spine length adjacent to the dendrite). Dendritic segments of at least 3 animals per group were morphologically analyzed resulting in 4010–8555 spines. To determine spine density, ~1500 μm of dendritic length was analyzed per experimental group. For the statistical analysis of synaptic plasticity (density and morphology), we used the Nested one-way ANOVA and Tukey's multiple comparisons test. Values of $P < 0.05$ were considered statistically significant. The analyses were performed using Prism 9.3.1 software (GraphPad, San Diego, CA, USA).

## Electrophysiology recordings

Acute brain slices were prepared as described previously (Wojtowicz and Mozrzymas, 2014). All of the experimental procedures were approved by the Local Ethics Committee. The animals were anesthetized with isoflurane and decapitated. The hippocampi were dissected and cut into 350-μm thick slices using a vibratome (VT1200S, Leica, Germany) in ice-cold buffer that contained 75 mM sucrose, 87 mM NaCl, 2.5 mM KCl, 1.25 mM $NaH_2PO_4$, 25 mM $NaHCO_3$, 0.5 mM $CaCl_2$, 10 mM $MgSO_4*7H_2O$, and 20 mM glucose, pH 7.4. Slices were subsequently allowed to

recover in the same solution (32 °C, 15 min) and stored until the end of the experiments in the oxygenated artificial cerebrospinal fluid (aCSF) that contained 125 mM NaCl, 25 mM $NaHCO_3$, 2.6 mM KCl, 1.25 mM $NaH_2PO_4$, 2.0 mM $CaCl_2$, and 20 mM glucose, pH 7.4. All solutions were oxygenated with carbogen (95% $O_2$, 5% $CO_2$). Recordings were made in aCSF after 2 h of slice recovery. Schaeffer collateral axons were stimulated with a concentric bipolar electrode (0.1 Hz, 0.3 ms). Compound AMPAR- and NMDAR-mediated fEPSPs were recorded with glass micropipettes that were filled with aCSF (1–3 M Ω resistance) in the stratum radiatum of the CA1 region (150–200 μm from the stratum pyramidale). NMDAR-mediated signals were isolated from compound fEPSPs with the AMPA/kainate receptor antagonist DNQX (20 μM) and L-type calcium channel blocker nifedipine (20 μM) in $Mg^{2+}$-free solutions, as described previously (Wojtowicz and Mozrzymas, 2014). At the end of each recording, the NMDAR antagonist APV (50 μM) was used to confirm the origin of the recorded fEPSPs. All of the drugs were obtained from Sigma-Aldrich (Poland) and Tocris (UK). The electrophysiology data were analyzed using pClamp10.7.0.3 software (Molecular Devices, USA) and AxoGraphX software (developed by John Clements) as described previously (Wojtowicz and Mozrzymas, 2014). Recordings were performed blinded to the genotype group.

## Serial block-face scanning electron microscopy (SBEM)

The mice were transcardially perfused with 2% PFA (P6148 Sigma-Aldrich) with 2% glutaraldehyde (GA, EM grade, G5882 Sigma-Aldrich) in 0.1 M phosphate buffer pH 7.4 in dd$H_2O$. The brains were gently dissected from the skull and fixed for SBEM overnight at 4 °C. Next, the brains were cut into 100 μm slices, and the hippocampi were dissected for the staining (blade travel speed: 0.075 mm/s, cutting frequency: 80 Hz). The SBEM staining was performed according to previously published protocol (Corpus ID: 136219796). Slices were postfixed with solution of 2% osmium tetroxide (#75632 Sigma-Aldrich) and 1.5% potassium ferrocyanide (P3289 Sigma-Aldrich) in 0.1 M phosphate buffer pH 7.4 for 60 min on ice. Next, samples were rinsed 5 × 3 min with dd$H_2O$ and subsequently exposed to 1% aqueous thiocarbohydrazide TCH (#88535 Sigma-Aldrich) solution for 20 min. Samples were then washed 5 × 3 min with ddH2O and stained with osmium tetroxide (1% Osmium Tetroxide in dd$H_2O$) for 30 min RT. Afterward, slices were rinsed 5 × 3 min with ddH2O and incubated in 1% aqueous solution of uranyl acetate overnight in 4 °C. The next day, lead aspartate solution was prepared by dissolving lead nitrate (0.066 g) in 10 ml L-aspartic acid (0.998 g of L-aspartic acid (Sigma-Aldrich) in 250 ml of dd$H_2O$). Slices were rinsed 5 × 3 min with ddH2O, incubated with lead aspartate for 30 min in 60 °C and then washed 5 × 3 min with degassed (autoclaved) ddH2O and dehydration was performed using graded dilutions of ethanol (ice-cold for better membrane preservation, 30%, 50%, 70%, 80%, 90%, and 2 × 100% ethanol, 5 min each). Samples were infiltrated with a resin that was prepared by mixing: A (17 g), B (17 g) and D (0.51 g) components of Durcupan (#44610 Sigma-Aldrich) with 8 drops of DMP-30 (#45348 Sigma-Aldrich) accelerator (Knott et al, 2009). Part of the resin was then mixed 1:1 (v/v) with 100% ethanol and slices were incubated in 50% resin for 30 min in RT. The resin was then replaced with 100% Durcupan for 1 h in RT and then 100% Durcupan infiltration was performed o/n. The next day, samples

were infiltrated with freshly prepared resin (as described above) for another 2 h in RT, then flat embedded between Aclar sheets (Ted Pella #10501-10). Samples were put in a laboratory oven for at least 48 h, 65–70 °C—for the resin to polymerize. After resin hardening, Aclar layers were separated and the resin-embedded samples were taken out. Squares, of ~1 mm × 1 mm, cut out with razor-blades, were attached to aluminum pins (Gatan metal rivets, Oxford instruments) with a very small amount of cyanacrylate glue and then mounted to the ultramicrotome (Leica ultracut R) and trimmed. Samples were grounded with conductive silver paint (Ted Pella, 16062-15) to the pin and mounted into the 3 View chamber.

## 3 View imaging, scan processing, image analysis, and statistical analysis of the data

Samples were imaged with SigmaVP (Zeiss) scanning electron microscope equipped with 3 View 2 chamber using a backscatter electron detector. Scans were taken in the middle part of the stratum radiatum of the CA1 of the dorsal hippocampus. From each sample, 200 sections were collected (thickness 60 nm). Imaging settings: variable pressure18Pa, EHT 4 kV, aperture: 15 μm, pixel dwell time: 7 μs, pixel size: 5 nm (2048 × 2048 resolution). Obtained scans were aligned using the ImageJ software (ImageJ-> Plugins-> Registration-> StackReg) and saved as.tiff image sequence. Then the image sequences were imported to the Reconstruct software (Fiala, 2005) (http://synapses.clm.utexas.edu/tools/reconstruct/reconstruct.stm). Mitochondria density was analyzed with the modified unbiased brick method (Fiala and Harris, 2001) per tissue volume. For each sample, all mitochondria were counted in 1 brick. Size of each brick was 6.74 μm × 6.74 μm × 9.12 μm. A structure was considered to be a synaptic mitochondria when it was in a presynaptic site containing vesicles opposing postsynaptic site with electron-dense material. Image acquisition and analysis was performed blinded to the genotype group. Statistical tests were performed in Graphpad Prism 9.5.0. Details of statistical tests are provided in figure legends. Mitochondria volume and area followed normal distributions and were compared with Nested one-way ANOVA test. Mitochondria density followed normal distribution and was compare with ordinary one-way ANOVA, pairs of results were compared with Tukey's multiple comparisons test. Differences between groups were considered significant if $P < 0.05$. Mean values and standard errors of the mean (SEM) are shown.

## Preparation of synaptoneurosomes and MitoPlate™ assay

Synaptoneurosomes were prepared as described previously (Kuzniewska et al, 2018). Before tissue dissection, Krebs buffer (2.5 mM $CaCl_2$, 1.18 mM $KH_2PO_4$, 118.5 mM NaCl, 24.9 mM $NaHCO_3$, 1.18 mM $MgSO_4$, 3.8 mM $MgCl_2$, 212.7 mM glucose) was aerated with an aquarium pump for 30 min at 4 °C. Next, the pH was lowered to 7.4 using dry ice. The buffer was supplemented with 1×protease inhibitor cocktail cOmplete EDTA-free (Roche) and 40 U/ml RNase Inhibitor (RiboLock, Thermo Fisher Scientific). Animals were euthanized by cervical dislocation, hippocampi and a part of the cortex adjacent to the hippocampus were dissected. Tissue from one hemisphere (~50 mg) was homogenized in 1.5 ml Krebs buffer using a Dounce homogenizer with 10–12 strokes. All

steps were kept ice-cold to prevent stimulation of synaptoneurosomes. Homogenates were loaded into 20 ml syringe and passed through a series of pre-soaked (with Krebs buffer) nylon mesh filters consecutively 100, 60, 30, and 10 μm (Merck Millipore) in cold room to 50 ml polypropylene tube, centrifuged at 1000 × g for 15 min at 4 °C, washed and pellet was resuspended in Krebs buffer with protease and RNase inhibitors. Mitochondrial function assays using MitoPlates S1 (Biolog, Cat. #14105) were performed as suggested by the manufacturer. Briefly, Assay Mix was dispensed into all wells of a MitoPlate and the plate was incubated at 37 °C for 1 h to allow substrates to fully dissolve. Freshly isolated synaptoneurosomes were pelleted and resuspended in 1 × Biolog Mitochondrial Assay Solution (MAS). SN suspension was dispensed into each well of a MitoPlate (30 μl/well) and the color formation at 590 nm was read kinetically for 2 h on a Multiskan FC Microplate Photometer (Thermo Scientific). The background was corrected for the blank sample and the average rate between 10 and 100 min was calculated. The protein content of synaptoneurosomal samples was measured using Bradford method. Synaptoneurosomes isolated from three male and three female mice per genotype were analyzed. Results are presented as the average rate per minute per microgram of protein.

## Isolation of functional mitochondria

Mitochondria were isolated using discontinuous Percoll density gradient centrifugation as previously described (Ferreira et al, 2018). Briefly, hippocampi and cortices were dissected from mouse brains, washed once in ice-cold mitochondria isolation buffer (MIB: 225 mM mannitol, 75 mM sucrose, 1 mM EGTA, 5 mM HEPES/KOH, pH 7.2), transferred to a 1 ml Dounce-type tissue grinder (WHEATON® #357538) and homogenized in MIB supplemented with 1 mg/ml fatty acid-free BSA. 0.74 mL of MIB was used for 2 hippocampi or for cortex from one hemisphere. Tissue was homogenized 8 times using a pestle with 0.114 + /−0.025 mm clearance, followed by another 8 strokes with a pestle with 0.05 mm ± 0.025 mm clearance. The final homogenate was then centrifuged at 1100×g for 2 min, at 4 °C. The supernatant was collected and carefully mixed with freshly made 80% Percoll (GE Healthcare, catalog no. 17-5445-02) prepared in mitochondrial dilution buffer (MDB: 1000 mM sucrose, 50 mM HEPES/KOH, 10 mM EGTA, pH 7.0), to create a 9.5% Percoll solution, which was further carefully layered on the top of freshly made 10% Percoll (diluted from 80% Percoll in MIB). The mitochondrial fraction was pelleted by centrifugation at 18,500×g for 10 min at 4 °C. The pellet was then resuspended in 1 mL of mitochondria washing buffer (MWB: 250 mM sucrose, 5 mM HEPES/KOH, 0.1 mM EGTA, pH 7.2) and centrifuged at 10,000×g for 5 min at 4 °C. Mitochondrial pellets from hippocampi and cortices were resuspended and pooled together in final 80 μl of ice-cold MWB, to create a concentrated mitochondria solution and immediately used to measure respiration. The protein concentration of isolated mitochondria was quantified using the Bradford assay.

## High-resolution respirometry

$O_2$ consumption rates in isolated brain mitochondria were measured polarographically using a high-resolution respirometer (Oroboros Oxygraph-O2K). In total, 70 μl of freshly isolated

mitochondria (~3 mg/ml) were loaded per respirator chamber. Respiration was measured at 37 °C in mitochondria assay solution (MAS: 70 mM sucrose, 220 mM mannitol, 10 mM $K_2HPO_4$, 5 mM $MgCl_2$, 1 mM EGTA, 2 mM HEPES/KOH) supplemented with 0.2% (w/v) fatty acid-free BSA as well as 10 mM pyruvate, 2 mM malate and 2.5 mM glutamate to support electron entry through complex I). Next CCCP was added to a final concentration of 0.5 μM. After inhibition of complex I by rotenone (2.5 μM), succinate (2.5 mM) was added to support electron entry through complex II. Next, complex III was inhibited by antimycin A (1 μl/ml) and ascorbate/TMPD (1 M/10 mM) were added to feed complex IV. Simultaneously, protein concentration was measured using the Bradford method and the obtained results were recalculated to mg of protein. Data recording was performed using Oxygraph-O2k and analyzed with DatLab 7 software (Oroboros Instruments).

## Graphics

Graphical abstract, Figs. 3F and EV3C,D were created in part using BioRender.com.

## Statistics

All the statistical details including statistical tests used and number of biological replicates are described in the respective paragraphs in "Methods" and in the Figure Legends. The list of statistical methods used and $P$ values is provided in the Appendix Table S1. The data points were subjected to the Shapiro–Wilk normality test or D'Agostino and Pearson normality test. We did not use any statistical test to predetermine the sample size. The sample size was chosen on the basis of our experience and good laboratory practice. The experimenter was blind to the animals' genotypes during the behavioral observations and video analysis.

## Data availability

All data are available in the main text or the supplementary materials. The whole-exome sequencing data that support the findings of this study are available on request from the corresponding author RP. The data are not publicly available due to ethical restriction (data contain information that could compromise the privacy of research participants). The datasets produced in this study are available in the following databases: [RNA-Sequencing data]: [NCBI's Gene Expression Omnibus] [GSE226319].

The source data of this paper are collected in the following database record: biostudies:S-SCDT-10_1038-S44321-024-00147-6.

## Peer review information

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

### The paper explained

**Problem**
Autism spectrum disorders (ASD) is a group of early-onset neurodevelopmental disorders ~4–5-fold more prevalent in males than females, however the mechanism underlying the sex bias of autism remains elusive. ASD and mitochondrial disorders share common clinical features, and the prevalence of mitochondrial disease is 500 times higher in the ASD population, pointing to the role of mitochondrial homeostasis in autism, yet the causal relationships are unclear.

**Results**
We identified a point mutation in the HSP90 family mitochondrial chaperone, TRAP1 p.Q639* in two unrelated male ASD patients. We generated a corresponding knock-in mouse model, that displayed social behavior abnormalities. We identified altered dendritic spine morphology and basal synaptic neurotransmission, fewer presynaptic mitochondria and changes in mitochondrial metabolism in the brains of Trap1 mutant mice. Interestingly, observed symptoms were more pronounced in males than in females.

**Impact**
Here we provide the first example of a monogenic ASD caused by impaired mitochondrial protein homeostasis, namely TRAP1 p.Q639* mutation. Our study emphasizes the need for comprehensive analysis of both male and females in mouse models of ASD. Our work supports the potential role of mitochondria in autism etiology and suggests that exploring strategies to modulate mitochondrial metabolism could be promising for developing effective treatments for ASD.

Cibulskis K, Lawrence MS, Carter SL, Sivachenko A, Jaffe D, Sougnez C, Gabriel S, Meyerson M, Lander ES, Getz G (2013) Sensitive detection of somatic point mutations in impure and heterogeneous cancer samples. Nat Biotechnol 31:213–219

Demarest TG, Schuh RA, Waddell J, McKenna MC, Fiskum G (2016) Sex-dependent mitochondrial respiratory impairment and oxidative stress in a rat model of neonatal hypoxic-ischemic encephalopathy. J Neurochem 137:714–729

Devlin B, Scherer SW (2012) Genetic architecture in autism spectrum disorder. Curr Opin Genet Dev 22:229–237

D'Gama AM (2021) Somatic mosaicism and autism spectrum disorder. Genes 12:1699

D'Gama AM, Pochareddy S, Li M, Jamuar SS, Reiff RE, Lam AN, Sestan N, Walsh CA (2015) Targeted DNA sequencing from autism spectrum disorder brains implicates multiple genetic mechanisms. Neuron 88:910–917

Dou Y, Yang X, Li Z, Wang S, Zhang Z, Ye AY, Yan L, Yang C, Wu Q, Li J, Zhao B, Huang AY, Wei L (2017) Postzygotic single-nucleotide mosaicisms contribute to the etiology of autism spectrum disorder and autistic traits and the origin of mutations. Hum Mutat 38:1002–1013

Erlich Y (2011) Blood ties: chimerism can mask twin discordance in high-throughput sequencing. Twin Res Hum Genet 14:137–143

Fassio A, Patry L, Congia S, Onofri F, Piton A, Gauthier J, Pozzi D, Messa M, Defranchi E, Fadda M, Corradi A, Baldelli P, Lapointe L, St-Onge J, Meloche C, Mottron L, Valtorta F, Khoa Nguyen D, Rouleau GA, Benfenati F et al (2011) SYN1 loss-of-function mutations in autism and partial epilepsy cause impaired synaptic function. Hum Mol Genet 20:2297–2307

Ferreira IL, Carmo C, Naia L, Mota SI, Cristina Rego A (2018) Assessing mitochondrial function in in vitro and ex vivo models of Huntington's disease. Methods Mol Biol 1780:415–442

Ferri SL, Abel T, Brodkin ES (2018) Sex differences in autism spectrum disorder: a review. Curr Psychiatry Rep 20:9

Fiala JC (2005) Reconstruct: a free editor for serial section microscopy. J Microsc 218:52–61

Fiala JC, Harris KM (2001) Extending unbiased stereology of brain ultrastructure to three-dimensional volumes. J Am Med Inform Assoc 8:1–16

Filice F, Janickova L, Henzi T, Bilella A, Schwaller B (2020) The parvalbumin hypothesis of autism spectrum disorder. Front Cell Neurosci 14:577525

Fitzgerald JC, Zimprich A, Carvajal Berrio DA, Schindler KM, Maurer B, Schulte C, Bus C, Hauser AK, Kubler M, Lewin R, Bobbili DR, Schwarz LM, Vartholomaiou E, Brockmann K, Wust R, Madlung J, Nordheim A, Riess O, Martins LM, Glaab E et al (2017) Metformin reverses TRAP1 mutation-associated alterations in mitochondrial function in Parkinson's disease. Brain 140:2444–2459

Forsberg LA, Gisselsson D, Dumanski JP (2017) Mosaicism in health and disease—clones picking up speed. Nat Rev Genet 18:128–142

Freed D, Pevsner J (2016) The contribution of mosaic variants to autism spectrum disorder. PLoS Genet 12:e1006245

Gaugler T, Klei L, Sanders SJ, Bodea CA, Goldberg AP, Lee AB, Mahajan M, Manaa D, Pawitan Y, Reichert J, Ripke S, Sandin S, Sklar P, Svantesson O, Reichenberg A, Hultman CM, Devlin B, Roeder K, Buxbaum JD (2014) Most genetic risk for autism resides with common variation. Nat Genet 46:881–885

Greco B, Manago F, Tucci V, Kao HT, Valtorta F, Benfenati F (2013) Autism-related behavioral abnormalities in synapsin knockout mice. Behav Brain Res 251:65–74

Hollis F, Kanellopoulos AK, Bagni C (2017) Mitochondrial dysfunction in autism spectrum disorder: clinical features and perspectives. Curr Opin Neurobiol 45:178–187

Huang GZ, Woolley CS (2012) Estradiol acutely suppresses inhibition in the hippocampus through a sex-specific endocannabinoid and mGluR-dependent mechanism. Neuron 74:801–808

Jamuar SS, Lam AT, Kircher M, D'Gama AM, Wang J, Barry BJ, Zhang X, Hill RS, Partlow JN, Rozzo A, Servattalab S, Mehta BK, Topcu M, Amrom D, Andermann E, Dan B, Parrini E, Guerrini R, Scheffer IE, Berkovic SF et al (2014) Somatic mutations in cerebral cortical malformations. New Engl J Med 371:733–743

Knott GW, Holtmaat A, Trachtenberg JT, Svoboda K, Welker E (2009) A protocol for preparing GFP-labeled neurons previously imaged in vivo and in slice preparations for light and electron microscopic analysis. Nat Protoc 4:1145–1156

Krupp DR, Barnard RA, Duffourd Y, Evans SA, Mulqueen RM, Bernier R, Riviere JB, Fombonne E, O'Roak BJ (2017) Exonic mosaic mutations contribute risk for autism spectrum disorder. Am J Hum Genet 101:369–390

Kuzniewska B, Chojnacka M, Milek J, Dziembowska M (2018) Preparation of polysomal fractions from mouse brain synaptoneurosomes and analysis of polysomal-bound mRNAs. J Neurosci Methods 293:226–233

Landrum MJ, Lee JM, Riley GR, Jang W, Rubinstein WS, Church DM, Maglott DR (2014) ClinVar: public archive of relationships among sequence variation and human phenotype. Nucleic Acids Res 42:D980–5

Lim CS, Kim MJ, Choi JE, Islam MA, Lee YK, Xiong Y, Shim KW, Yang JE, Lee RU, Lee J, Park P, Kwak JH, Seo H, Kim CH, Lee JH, Lee YS, Hwang SK, Lee K, Lee JA, Kaang BK (2021) Dysfunction of NMDA receptors in neuronal models of an autism spectrum disorder patient with a DSCAM mutation and in Dscam-knockout mice. Mol Psychiatry 26:7538–7549

Lim ET, Uddin M, De Rubeis S, Chan Y, Kamumbu AS, Zhang X, D'Gama AM, Kim SN, Hill RS, Goldberg AP, Poultney C, Minshew NJ, Kushima I, Aleksic B, Ozaki N, Parellada M, Arango C, Penzol MJ, Carracedo A, Kolevzon A et al (2017) Rates, distribution and implications of postzygotic mosaic mutations in autism spectrum disorder. Nat Neurosci 20:1217–1224

Lin HC, Gean PW, Wang CC, Chan YH, Chen PS (2013) The amygdala excitatory/inhibitory balance in a valproate-induced rat autism model. PLoS ONE 8:e55248

Lisanti S, Tavecchio M, Chae YC, Liu Q, Brice AK, Thakur ML, Languino LR, Altieri DC (2014) Deletion of the mitochondrial chaperone TRAP-1 uncovers global reprogramming of metabolic networks. Cell Rep 8:671–677

Lord C, Risi S, Lambrecht L, Cook Jr. EH, Leventhal BL, DiLavore PC, Pickles A, Rutter M (2000) The autism diagnostic observation schedule-generic: a standard measure of social and communication deficits associated with the spectrum of autism. J Autism Dev Disord 30:205–223

Luo W, Zhang C, Jiang YH, Brouwer CR (2018) Systematic reconstruction of autism biology from massive genetic mutation profiles. Sci Adv 4:e1701799

Magnowska M, Gorkiewicz T, Suska A, Wawrzyniak M, Rutkowska-Wlodarczyk I, Kaczmarek L, Wlodarczyk J (2016) Transient ECM protease activity promotes synaptic plasticity. Sci Rep 6:27757

Martinez-Cerdeno V (2017) Dendrite and spine modifications in autism and related neurodevelopmental disorders in patients and animal models. Dev Neurobiol 77:393–404

Michaluk P, Wawrzyniak M, Alot P, Szczot M, Wyrembek P, Mercik K, Medvedev N, Wilczek E, De Roo M, Zuschratter W, Muller D, Wilczynski GM, Mozrzymas JW, Stewart MG, Kaczmarek L, Wlodarczyk J (2011) Influence of matrix metalloproteinase MMP-9 on dendritic spine morphology. J Cell Sci 124:3369–3380

Monteiro P, Feng G (2017) SHANK proteins: roles at the synapse and in autism spectrum disorder. Nat Rev Neurosci 18:147–157

Munch C, Harper JW (2016) Mitochondrial unfolded protein response controls matrix pre-RNA processing and translation. Nature 534:710–713

Palmieri L, Papaleo V, Porcelli V, Scarcia P, Gaita L, Sacco R, Hager J, Rousseau F, Curatolo P, Manzi B, Militerni R, Bravaccio C, Trillo S, Schneider C, Melmed R, Elia M, Lenti C, Saccani M, Pascucci T, Puglisi-Allegra S et al (2010) Altered calcium homeostasis in autism-spectrum disorders: evidence from

biochemical and genetic studies of the mitochondrial aspartate/glutamate carrier AGC1. Mol Psychiatry 15:38–52

Peca J, Feliciano C, Ting JT, Wang W, Wells MF, Venkatraman TN, Lascola CD, Fu Z, Feng G (2011) Shank3 mutant mice display autistic-like behaviours and striatal dysfunction. Nature 472:437–442

Phillips M, Pozzo-Miller L (2015) Dendritic spine dysgenesis in autism related disorders. Neurosci Lett 601:30–40

Purushottam Dharaskar S, Paithankar K, Kanugovi Vijayavittal A, Shabbir Kara H, Amere Subbarao S (2020) Mitochondrial chaperone, TRAP1 modulates mitochondrial dynamics and promotes tumor metastasis. Mitochondrion 54:92–101

Puscian A, Leski S, Kasprowicz G, Winiarski M, Borowska J, Nikolaev T, Boguszewski PM, Lipp HP, Knapska E (2016) Eco-HAB as a fully automated and ecologically relevant assessment of social impairments in mouse models of autism. eLife 5:e19532

Ramoz N, Reichert JG, Smith CJ, Silverman JM, Bespalova IN, Davis KL, Buxbaum JD (2004) Linkage and association of the mitochondrial aspartate/glutamate carrier SLC25A12 gene with autism. Am J Psychiatry 161:662–669

Rangaraju V, Lauterbach M, Schuman EM (2019) Spatially stable mitochondrial compartments fuel local translation during plasticity. Cell 176:73–84 e15

Regehr WG (2012) Short-term presynaptic plasticity. Cold Spring Harb Perspect Biol 4:a005702

Reuter MS, Tawamie H, Buchert R, Hosny Gebril O, Froukh T, Thiel C, Uebe S, Ekici AB, Krumbiegel M, Zweier C, Hoyer J, Eberlein K, Bauer J, Scheller U, Strom TM, Hoffjan S, Abdelraouf ER, Meguid NA, Abboud A, Al Khateeb MA et al (2017) Diagnostic yield and novel candidate genes by exome sequencing in 152 consanguineous families with neurodevelopmental disorders. JAMA Psychiatry 74:293–299

Robinson JT, Thorvaldsdottir H, Winckler W, Guttman M, Lander ES, Getz G, Mesirov JP (2011) Integrative genomics viewer. Nat Biotechnol 29:24–26

Rutter M, Le Couteur A, Lord C (2003) Autism diagnostic interview-revised. Western Psychol Services 29:30

Rydzanicz M, Glinkowski W, Walczak A, Koppolu A, Kostrzewa G, Gasperowicz P, Pollak A, Stawinski P, Ploski R (2022) Postzygotic mosaicism of a novel PTPN11 mutation in monozygotic twins discordant for metachondromatosis. Am J Med Genet A 188:1482–1487

Rydzanicz M, Olszewski P, Kedra D, Davies H, Filipowicz N, Bruhn-Olszewska B, Cavalli M, Szczaluba K, Mlynek M, Machnicki MM, Stawinski P, Kostrzewa G, Krajewski P, Sladowski D, Chrzanowska K, Dumanski JP, Ploski R (2021) Variable degree of mosaicism for tetrasomy 18p in phenotypically discordant monozygotic twins-diagnostic implications. Mol Genet Genomic Med 9:e1526

Sacai H, Sakoori K, Konno K, Nagahama K, Suzuki H, Watanabe T, Watanabe M, Uesaka N, Kano M (2020) Autism spectrum disorder-like behavior caused by reduced excitatory synaptic transmission in pyramidal neurons of mouse prefrontal cortex. Nat Commun 11:5140

Silverman JL, Yang M, Lord C, Crawley JN (2010) Behavioural phenotyping assays for mouse models of autism. Nat Rev Neurosci 11:490–502

Stenson PD, Ball EV, Mort M, Phillips AD, Shiel JA, Thomas NS, Abeysinghe S, Krawczak M, Cooper DN (2003) Human Gene Mutation Database (HGMD): 2003 update. Hum Mutat 21:577–581

Sudhof TC (2008) Neuroligins and neurexins link synaptic function to cognitive disease. Nature 455:903–911

Tabatadze N, Huang G, May RM, Jain A, Woolley CS (2015) Sex differences in molecular signaling at inhibitory synapses in the hippocampus. J Neurosci 35:11252–11265

Tabuchi K, Blundell J, Etherton MR, Hammer RE, Liu X, Powell CM, Sudhof TC (2007) A neuroligin-3 mutation implicated in autism increases inhibitory synaptic transmission in mice. Science 318:71–76

Takeuchi K, Gertner MJ, Zhou J, Parada LF, Bennett MV, Zukin RS (2013) Dysregulation of synaptic plasticity precedes appearance of morphological defects in a Pten conditional knockout mouse model of autism. Proc Natl Acad Sci USA 110:4738–4743

Tick B, Bolton P, Happe F, Rutter M, Rijsdijk F (2016) Heritability of autism spectrum disorders: a meta-analysis of twin studies. J Child Psychol Psychiatry 57:585–595

Uhl M, Schmeisser MJ, Schumann S (2022) The sexual dimorphic synapse: from spine density to molecular composition. Front Mol Neurosci 15:818390

Uzunova G, Hollander E, Shepherd J (2014) The role of ionotropic glutamate receptors in childhood neurodevelopmental disorders: autism spectrum disorders and fragile x syndrome. Curr Neuropharmacol 12:71–98

Vicari S, Napoli E, Cordeddu V, Menghini D, Alesi V, Loddo S, Novelli A, Tartaglia M (2019) Copy number variants in autism spectrum disorders. Prog Neuropsychopharmacol Biol Psychiatry 92:421–427

Werling DM (2016) The role of sex-differential biology in risk for autism spectrum disorder. Biol Sex Differ 7:58

Wickham H (2016) ggplot2: elegant graphics for data analysis. Springer-Verlag New York. https://ggplot2.tidyverse.org

Wojtowicz T, Mozrzymas JW (2014) Matrix metalloprotease activity shapes the magnitude of EPSPs and spike plasticity within the hippocampal CA3 network. Hippocampus 24:135–153

Woodbury-Smith M, Lamoureux S, Begum G, Nassir N, Akter H, O'Rielly DD, Rahman P, Wintle RF, Scherer SW, Uddin M (2022) Mutational landscape of autism spectrum disorder brain tissue. Genes 13:207

Yoo J, Bakes J, Bradley C, Collingridge GL, Kaang BK (2014) Shank mutant mice as an animal model of autism. Philos Trans R Soc Lond B Biol Sci 369:20130143

Yoshida S, Tsutsumi S, Muhlebach G, Sourbier C, Lee MJ, Lee S, Vartholomaiou E, Tatokoro M, Beebe K, Miyajima N, Mohney RP, Chen Y, Hasumi H, Xu W, Fukushima H, Nakamura K, Koga F, Kihara K, Trepel J, Picard D et al (2013) Molecular chaperone TRAP1 regulates a metabolic switch between mitochondrial respiration and aerobic glycolysis. Proc Natl Acad Sci USA 110:E1604–12

Youssoufian H, Pyeritz RE (2002) Mechanisms and consequences of somatic mosaicism in humans. Nat Rev Genet 3:748–758

Zoghbi HY, Bear MF (2012) Synaptic dysfunction in neurodevelopmental disorders associated with autism and intellectual disabilities. Cold Spring Harb Perspect Biol 4:a009886

## Acknowledgements

This work was funded by the National Science Centre (NCN) Poland grant number 2014/13/B/NZ5/00287 to MR, by NCN grant 2019/35/B/NZ4/04355 to MD, TEAM TECH CORE FACILITY/2017-4/5 to AD. The authors thank prof. K Radwanska, the Nencki Institute of Experimental Biology in Warsaw for granting access to the electrophysiology setup. NGS was performed thanks to Genomics Core Facility CeNT UW (RRID:SCR_022718), using NovaSeq 6000 platform financed by Polish Ministry of Science and Higher Education (decision no. 6817/IA/SP/2018 of 2018-04-10). IIMCB core facilities, the IIMCB IN-MOL-CELL Infrastructure, were funded by the European Union—NextGenerationEU under the National Recovery Plan, co-financed by the European Union under the European Funds for Smart Economy 2021-2027 (FENG) and funded by the European Union. Electron microscopy experiments were performed at the Laboratory of Imaging Tissue Structure and Function which serves as an imaging core facility at the Nencki Institute of Experimental Biology in Warsaw and is part of the infrastructure of the Polish Euro-BioImaging Node financed by Polish Ministry of Science and Higher Education contract no. 2022/WK/05; Polish Euro-BioImaging Node "Advanced Light Microscopy Node Poland"). 3-chamber

sociability task was performed at the Laboratory of Behavioral Methods which serves as a core facility at the Nencki Institute of Experimental Biology in Warsaw.

## Author contributions

**Małgorzata Rydzanicz**: Conceptualization; Funding acquisition; Investigation; Writing—original draft. **Bozena Kuzniewska**: Conceptualization; Formal analysis; Investigation; Visualization; Writing—original draft; Writing—review and editing. **Marta Magnowska**: Formal analysis; Investigation; Visualization. **Tomasz Wojtowicz**: Formal analysis; Investigation; Visualization. **Aleksandra Stawikowska**: Investigation. **Anna Hojka**: Data curation; Formal analysis; Visualization. **Ewa Borsuk**: Methodology. **Ksenia Meyza**: Data curation; Formal analysis. **Olga Gewartowska**: Methodology. **Jakub Gruchota**: Methodology. **Jacek Milek**: Investigation. **Patrycja Wardaszka**: Investigation. **Izabela Chojnicka**: Investigation. **Ludwika Kondrakiewicz**: Investigation. **Dorota Dymkowska**: Methodology. **Alicja Puścian**: Formal analysis; Visualization. **Ewelina Knapska**: Conceptualization; Data curation; Writing—review and editing. **Andrzej Dziembowski**: Conceptualization; Supervision; Funding acquisition; Writing—review and editing. **Rafał Ploski**: Conceptualization; Resources; Supervision. **Magdalena Dziembowska**: Conceptualization; Supervision; Funding acquisition; Writing—original draft; Project administration; Writing—review and editing.

Source data underlying figure panels in this paper may have individual authorship assigned. Where available, figure panel/source data authorship is listed in the following database record: biostudies:S-SCDT-10_1038-S44321-024-00147-6.

## Disclosure and competing interests statement

The authors declare no competing interests.

# Expanded View Figures

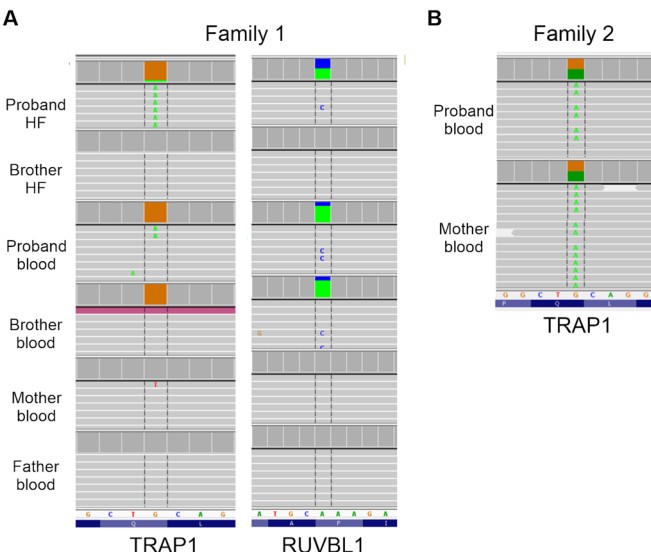

**Figure EV1.   NGS-based deep amplicon sequencing of postzygotic *TRAP1* and *RUVBL1* variants identified by whole-exome sequencing in a pair of ASD-discordant MZTs and the *TRAP1* p.Q639\* variant from a replication cohort individual.**

(**A**) Variants *TRAP1* p.Q639\* and *RUVBL1* p.F329L were verified in the ASD-affected twin and his ASD-unaffected twin brother in DNA samples purified from hair follicles (HF) and blood; parental analysis was done on blood samples only. In the ASD-affected twin the VAF of *TRAP1* p.Q639\* in HF DNA sample was 8% (genomic position coverage 13368×), and in the blood DNA sample, the VAF was 2% (genomic position coverage 27384x). In the unaffected brother HF DNA the variant was not present (genomic position coverage 7779x), while in the blood, the VAF was 2% (genomic position coverage 31972x). In the parent samples, only the wild-type sequence was identified (coverage 26714x for mother and 28222x for father). In the ASD-affected twin, the VAF of RUVBL1 p.F329L in HF DNA was 48% (coverage 54610×) and in the blood sample the VAF was 22% (coverage 49383×). In the unaffected brother's HF DNA only the wild-type sequence was identified (coverage 52237×), while in blood the VAF was 22% (coverage 40726×). In the parents, only the wild-type sequence was identified (coverage 57498× for mother and 51788× for father). (**B**) Verification of the heterozygous *TRAP1* p.Q639\* variant in an ASD patient from the replication cohort revealed inheritance from a ASD-unaffected mother (VAF 50% in both proband and mother); DNA from the proband's father was not available for testing. Deep amplicon sequencing results were viewed with the Integrative Genomics Viewer (IGV) tool.

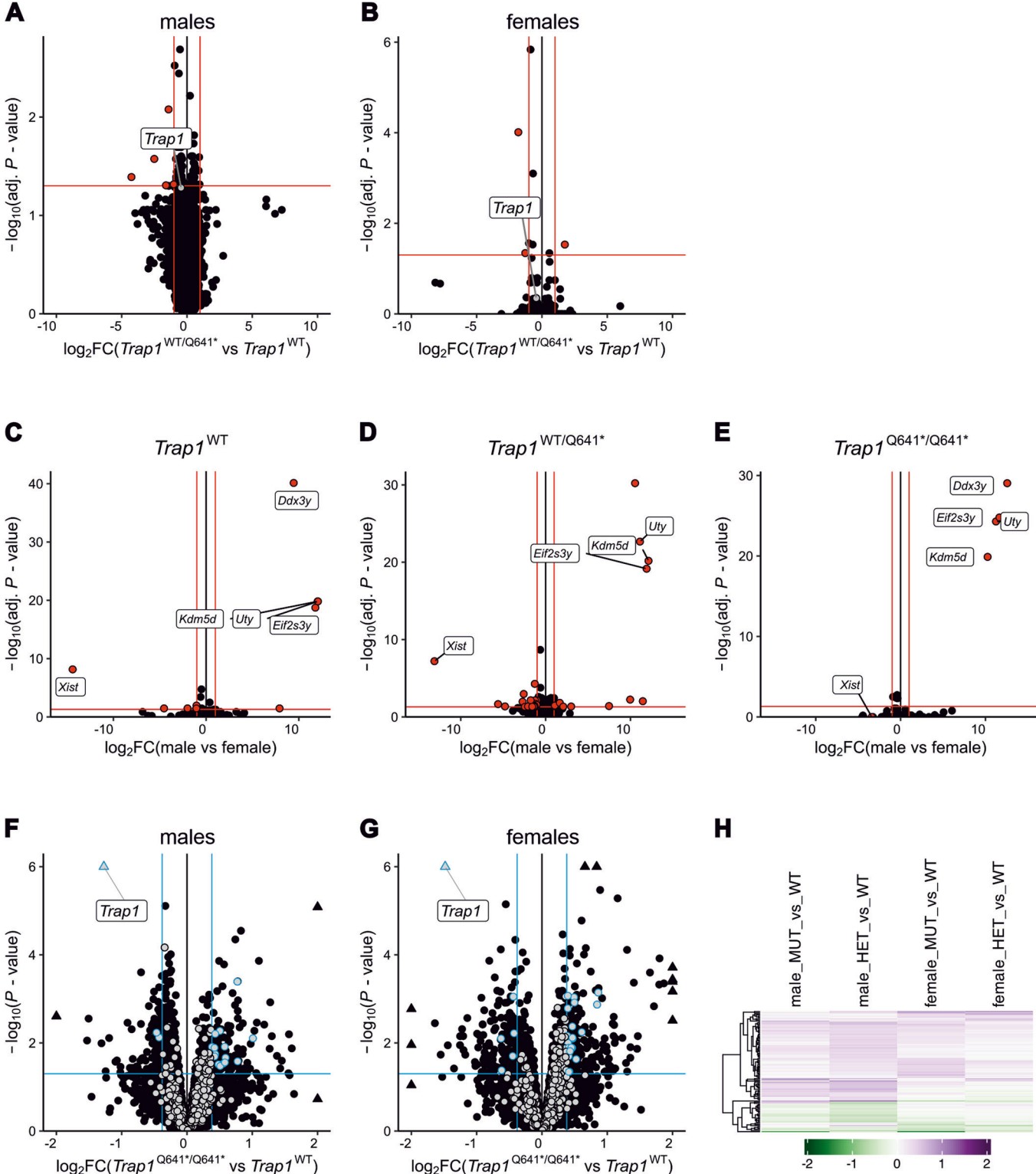

◀

**Figure EV2.  Transcriptional changes in *Trap1^{Q641*/Q641*}*, *Trap1^{WT/Q641*}* and *Trap1^{WT}* mice.**

(A–G) Volcano plots representing the global differential gene expression in RNA-Seq analysis of the hippocampi of *Trap1^{WT}*, *Trap1^{WT/Q641*}* and *Trap1^{Q641*/Q641*}* for male and female mice ($n = 3$–4 animals/group). (A–E) The x-axis indicates $\log_2$ fold changes ($\log_2$FC) of gene expression levels in *Trap1^{WT/Q641*}* versus *Trap1^{WT}* mice in males (A) and females (B) or changes in males versus females in *Trap1^{WT}* (C), *Trap1^{WT/Q641*}* (D) and *Trap1^{Q641*/Q641*}* (E). The y-axis indicates $-\log_{10}$ of adjusted P value (adj.P value). P values were calculated with Wald test statistics and were adjusted with Benjamini–Hochberg method. Black circles represent transcripts not differentially expressed, red circles represent transcripts significantly differentially expressed ($|\log2(FC)| >1$, adj.P value < 0.05–thresholds designated by red lines on the plot). The topmost differential genes are labeled by gene symbols. (F, G) Volcano plots representing the global differential gene expression in RNA-Seq analysis of the hippocampi of Trap1^{WT} and Trap1^{Q641*/Q641*} for male (F) and female (G) mice ($n = 3$–4 animals/group). The x-axis indicates log2 fold changes (log2FC) of gene expression levels in Trap1^{Q641*/Q641*} versus Trap1^{WT} mice, and the y-axis indicates –log10 of P value (not the adj.P value). Black circles represent transcripts not differentially expressed, gray circles represent genes coding proteins with mitochondrial localization, circles with blue outlines represent transcripts significantly differentially expressed ($|\log2(FC)| >0.38$, P value < 0.05–thresholds designated by blue lines on the plot). The triangle shapes on the plots are for the outlier transcripts that had values out of the scale presented on the plot and their values were displaced by the maximum/ minimum plotted value (changes applied on both axis). (H) Heatmap representation of expression changes of 103 genes coding proteins with mitochondrial localization on transcript levels from the RNA-Seq analysis. The genes were chosen if were significantly differentially expressed ($|\log2(FC)| >0.38$, P value < 0.05) in any of the plotted comparisons. The colors represent $\log_2$ of fold changes. Green represents genes which expression was lower in *Trap1^{Q641*/Q641*}* or *Trap1^{WT/Q641*}* and higher in *Trap1^{WT}*, whereas violet represents the opposite.

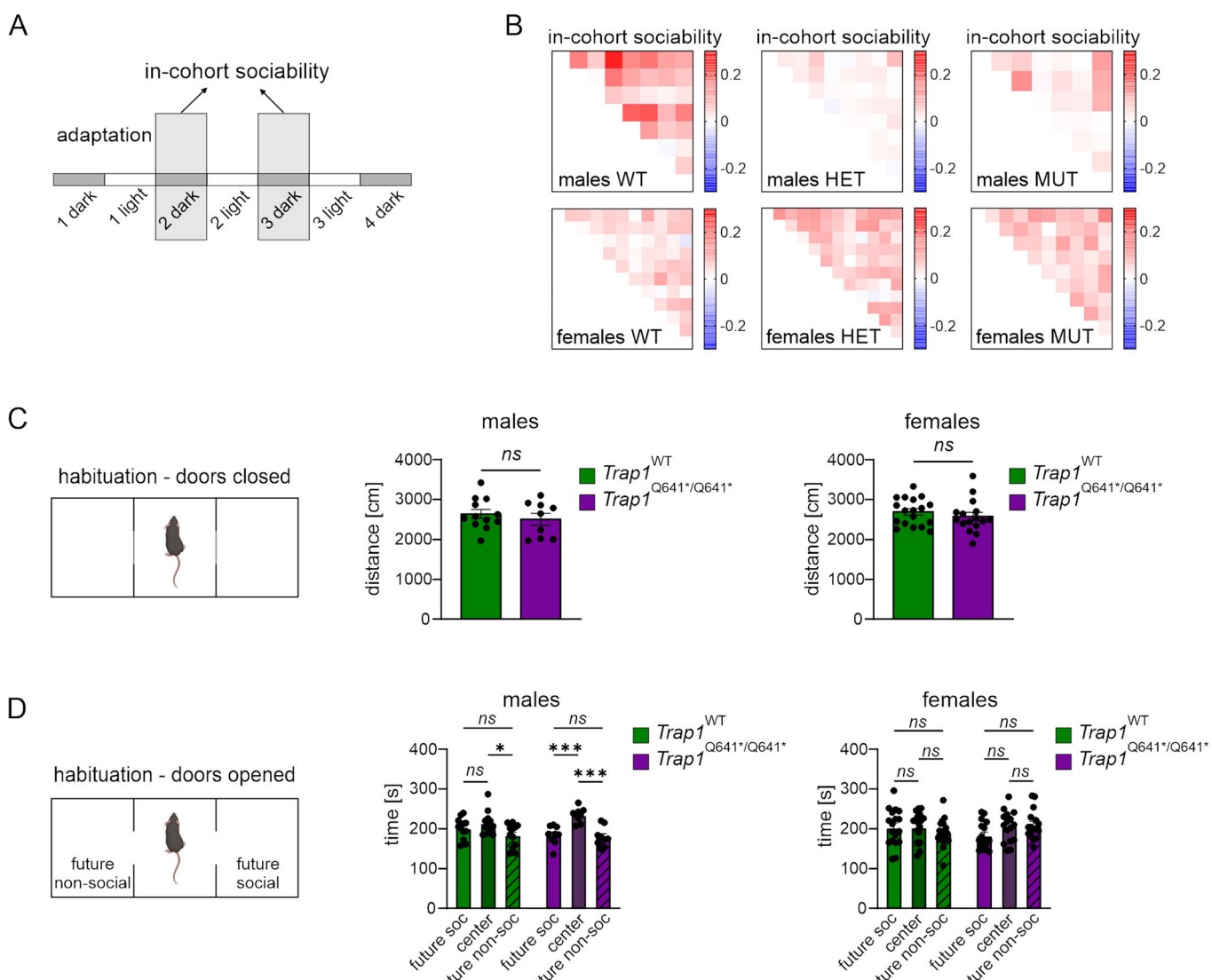

**Figure EV3. Eco-HAB and three-chamber behavioral analysis of *Trap1*^WT, *Trap1*^WT/Q641* and *Trap1*^Q641*/Q641* mice.**

(A, B) Eco-HAB behavioral analysis of *Trap1*^WT, *Trap1*^WT/Q641* and *Trap1*^Q641*/Q641* mouse cohorts. (A) Schematic depicting the experimental timeline and data used for analysis of Eco-HAB behavioral measures. Locomotor activity was assessed in 12-h bins and "in-cohort sociability" was calculated from the data collected during the 2nd and 3rd dark phase. (B) Heat maps depicting "in-cohort sociability" results. Each small square represents the "sociability" parameter for one pair of subjects, the color scale used is *blue* to *red*, with the *blue* representing "low sociability" and *red* "high sociability". $N = 7$–12 animals/group. Frequency distribution histograms for all pairs of animals from different cohorts are presented in main Fig. 3D,E. (C, D) Habituation phases of the three-chamber social approach task in *Trap1*^WT and *Trap1*^Q641*/Q641* mice. (C) Scheme showing the first habituation phase—with closed doors (left panel). The graphs show distance traveled by each mouse in the center chamber during 10 min. No differences in the locomotor activity were observed between the genotypes neither in males nor in females ($P > 0.05$, Mann–Whitney test; error bars indicate SEM). (D) Scheme showing the second habituation phase—with doors opened (left panel). The graphs show time spent by a mouse in each of the chambers during 10 min. Male *Trap1*^Q641*/Q641* mice preferred to spend more time in the already familiar center chamber as compared to the novel side chambers ($P < 0.0001$, two-way ANOVA, post hoc Tukey's multiple comparisons test; error bars indicate SEM). $N = 9$–12 and 16–18 animals/group for males and females respectively.

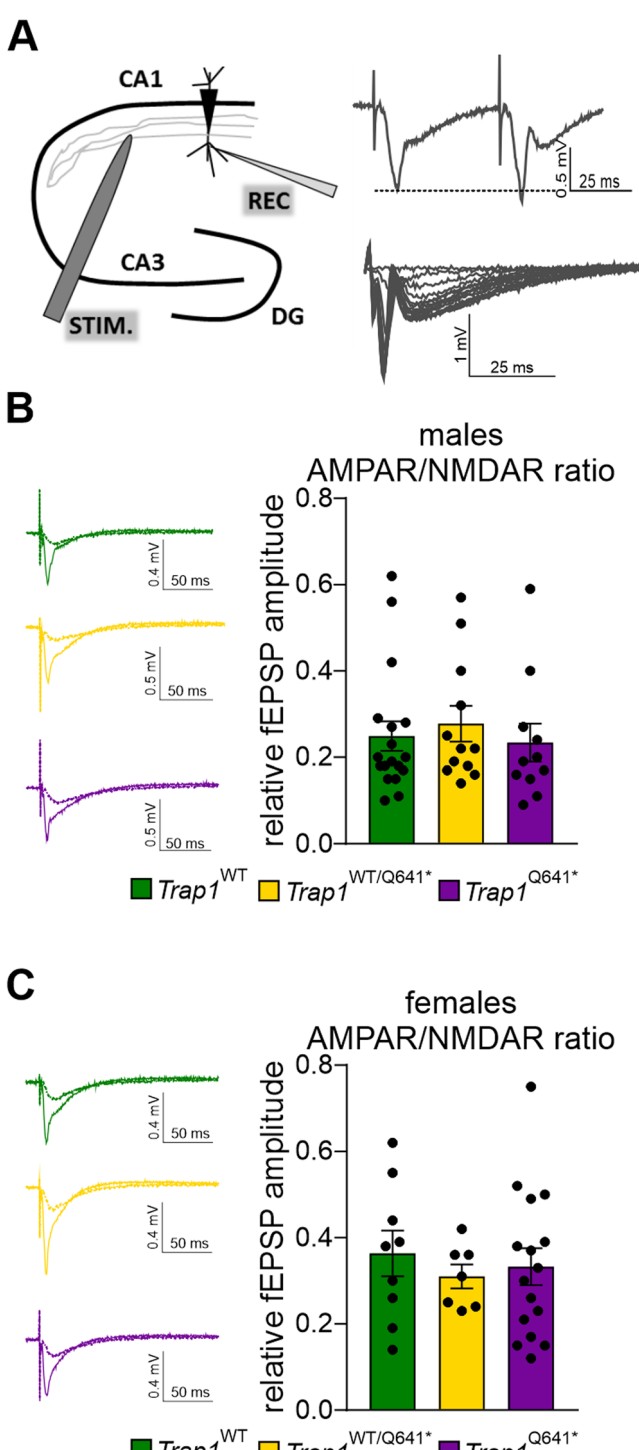

**Figure EV4.  Electrophysiological recordings of compound AMPAR- and NMDAR-mediated fEPSPs in the CA1 hippocampal region of *Trap1* mice.**

(A) Schematic of the electrophysiological recording setup depicting positions of stimulating (STIM) and recording (REC) electrodes in the CA1 hippocampal region. Top right, example trace of fEPSPs scaling in response to paired stimulation of Schaffer collaterals (interstimulus interval 50 ms). Bottom right, example traces of compound fEPSPs recorded in response to monotonically increasing stimuli applied to Schaeffer collaterals. (B) Quantification of changes in fEPSP amplitude following the application of the AMPAR antagonist DNQX. (20 μM). Sensitivity to DNQX and thus AMPAR/NMDAR ratio was not significantly different among male groups (one-way ANOVA, $P > 0.05$; error bars indicate SEM). (C) Quantification of changes in fEPSP amplitude following the application of the AMPAR antagonist DNQX. Sensitivity to DNQX was not significantly different among female groups (one-way ANOVA, $P > 0.05$; error bars indicate SEM). Insets in (B, C) show example recordings of compound fEPSPs before and after DNQX application. $N = 3$–6 animals, $n = 12$–25 slices (males); $N = 3$–4 animals, $n = 12$–17 slices (females).

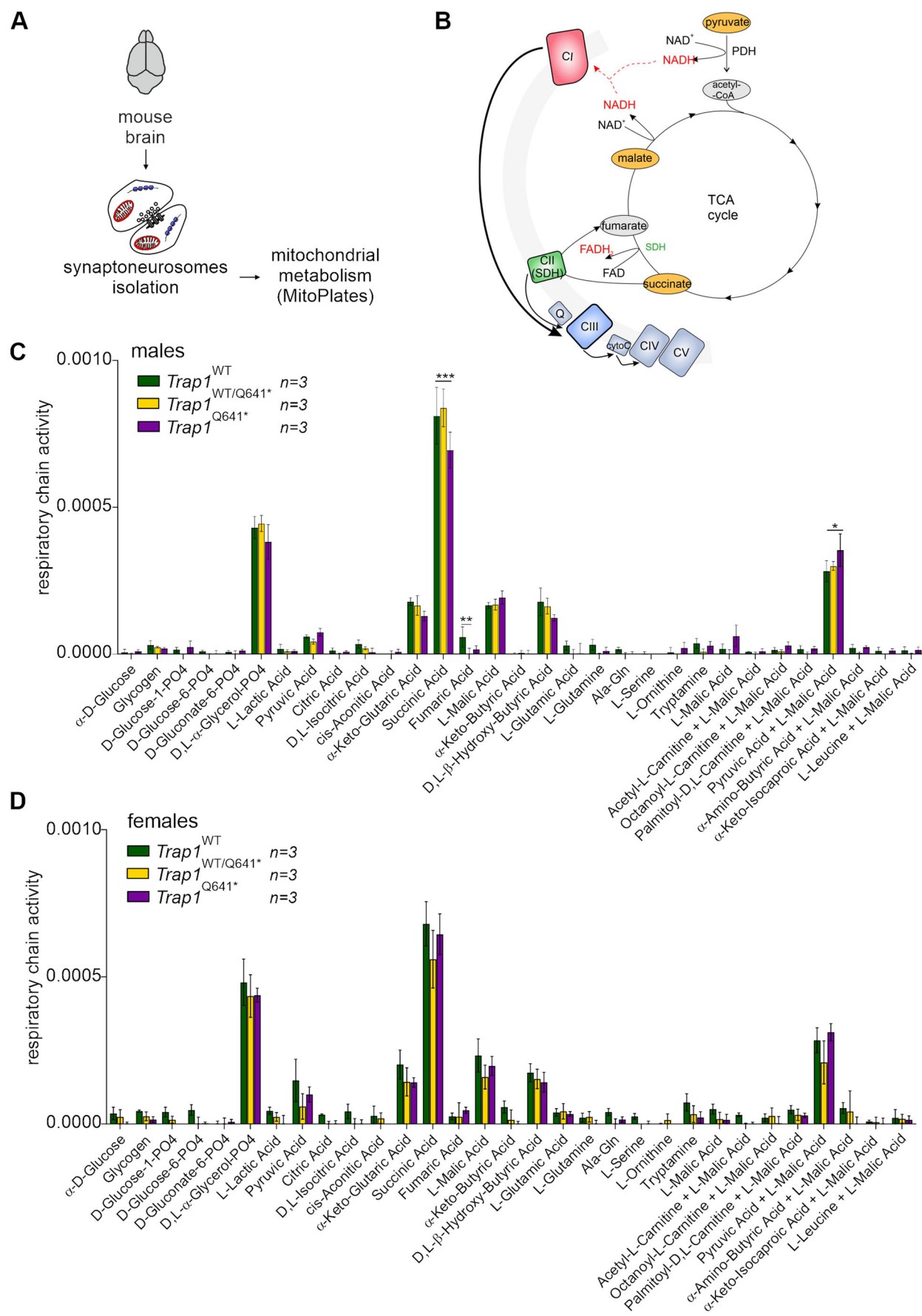

◀   **Figure EV5.   Functional mitochondrial phenotyping of synaptoneurosomes isolated from mouse brains (cortex and hippocampus) of male and female *Trap1* mice.**

The electron flow rates in the electron transport chain from 31 different bioenergetic substrates, including glycolysis, TCA cycle intermediates, fatty acids and amino acids, were measured using MitoPlates™. (**A**) Scheme showing the experimental workflow. (**B**) Scheme depicting selected substrate supply for mitochondrial respiration. Substrates differentially utilized in *Trap1*$^{Q641*/Q641*}$ synaptoneurosomes are marked in *yellow*. (**C**) In male *Trap1*$^{Q641*/Q641*}$ mice decreased usage of succinate (***$P < 0.001$) and fumarate (**$P < 0.01$) was observed. Also, increased consumption of pyruvate $+$malate (*$P < 0.05$) was noticed. (**D**) In contrast, in females no differences in usage of mitochondrial energy substrates were observed. Results are presented as the average rate/min/µg of protein, $+/-$ SEM ($n = 3$ per genotype/sex; two-way ANOVA, post hoc Sidak's multiple comparisons test).

