## [Peer Review File · EMBO Molecular Medicine]

Mutation in the mitochondrial chaperone TRAP1 leads to autism with more severe symptoms in males.

Małgorzata Rydzanicz, Bożena Kuzniewska, Marta Magnowska, Tomasz Wojtowicz, Aleksandra Stawikowska, Anna Hojka-Osinska, Ewa Borsuk, Ksenia Meyza, Olga Gewartowska, Jakub Gruchota, Jacek Milek, Patrycja Wardaszka, Izabela Chojnicka, Ludwika Kondrakiewicz, Dorota Dymkowska, Alicja Puścian, Ewelina Knapska, Andrzej Dziembowski, Rafał Płoski, and Magdalena Dziembowska

Corresponding authors: Magdalena Dziembowska (m.dziembowska@cent.uw.edu.pl) , Andrzej Dziembowski (adziembowski@iimcb.gov.pl)

Review Timeline:

Submission Date:	11th Dec 23
Editorial Decision:	25th Jan 24
Revision Received:	20th Aug 24
Editorial Decision:	2nd Sep 24
Revision Received:	13th Sep 24
Accepted:	16th Sep 24

Editor: Zeljko Durdevic

Transaction Report:

25th Jan 2024

Dear Dr. Dziembowska,

Thank you for the submission of your manuscript to EMBO Molecular Medicine, and please accept my apologies for the delay in getting back to you. We have now received feedback from two of the three reviewers who agreed to evaluate your manuscript. As the referee #2 will unfortunately not be able to return his/her report in a timely manner, and given that both reviewers provide very similar recommendations, we prefer to make a decision now in order to avoid further delay in the process.

As you will see from their reports pasted below, both referees recognize potential interest of the study but also raise serious and partially overlapping concerns that should be addressed in a major revision. Particular attention should be given to providing more mechanistic insight. Should referee #2 provide a report, we will send it to you, with the understanding that we will not ask for an additional revision. If you would like to discuss further the points raised by the referees, I am available to do so via email or video. Let me know if you are interested in this option.

We would welcome the submission of a revised version within three months for further consideration. Please let us know if you require longer to complete the revision.

I look forward to receiving your revised manuscript.

Yours sincerely,

Zeljko Durdevic

We require:

- 1) A .docx formatted version of the manuscript text (including legends for main figures, EV figures and tables). Please make sure that the changes are highlighted to be clearly visible.
- 2) Individual production quality figure files as .eps, .tif, .jpg (one file per figure). For guidance, download the 'Figure Guide PDF': (<https://www.embopress.org/page/journal/17574684/authorguide#figureformat>).
- 3) A .docx formatted letter INCLUDING the reviewers' reports and your detailed point-by-point responses to their comments. As part of the EMBO Press transparent editorial process, the point-by-point response is part of the Review Process File (RPF), which will be published alongside your paper.
- 4) A complete author checklist, which you can download from our author guidelines (<https://www.embopress.org/page/journal/17574684/authorguide#submissionofrevisions>). Please insert information in the checklist that is also reflected in the manuscript. The completed author checklist will also be part of the RPF.

6) It is mandatory to include a 'Data Availability' section after the Materials and Methods. Before submitting your revision, primary datasets produced in this study need to be deposited in an appropriate public database, and the accession numbers and database listed under 'Data Availability'. Please remember to provide a reviewer password if the datasets are not yet public (see <https://www.embopress.org/page/journal/17574684/authorguide#dataavailability>).

13) Author contributions: You will be asked to provide CRediT (Contributor Role Taxonomy) terms in the submission system. These replace a narrative author contribution section in the manuscript.

14) A Conflict of Interest statement should be provided in the main text.

Please also suggest a striking image or visual abstract to illustrate your article as a PNG file 550 px wide x 300-800 px high.

**** Reviewer's comments ****

Referee #1 (Comments on Novelty/Model System for Author):

It is an interesting finding but no mechanism.

Referee #1 (Remarks for Author):

In this paper, Rydzanicz et al. identified a novel mutation of p.Q639 in the TRAP1 gene from patients with ASD. Since the TRAP1 gene encodes a mitochondria chaperone of the HSP90 family, these findings support mitochondrial deficit theory in ASD pathogenesis. Moreover, the mice with this homo mutation show not only synaptic and mitochondrial abnormalities but also ASD-like symptoms only in males. Sex difference in ASD is also one of the most critical issues not revealed. Thus, the manuscript would be of interest to many readers. However, at least in the current manuscript, mechanistic insights such as male-specific ASD-like symptoms and mitochondria deficits caused by TRAP1 gene mutation were lacking.

1. In Figure 2D, the authors checked TRAP1 protein expression in mutant mice. However, it is unclear whether the point mutants p.Q639 generate truncated Trap1 protein. Therefore, the reviewer recommends the authors show a full-size WB image in a supplementary figure.

2. Related to #1, the authors should assess TRAP1 chaperon activity in the mutation.

3. Sex-specific differences in cellular and behavioral deficits in the Trap1 p.Q639 mice are key findings of the study. Thus, assessing basal protein and mRNA expression levels of each genotype and sex is indispensable.

4. In Figures 2F and G, the results of this expression analysis could not explain why TRAP1 point mutant causes male-specific ASD-like phenotypes as well as cellular-level deficits. The reviewer recommends the authors try a milder threshold. A proper threshold generates more DEGs, and following enrichment analysis may help to understand the core deficits of mutant mice from the viewpoint of gene expression.

5. The authors should show the list of RNA-sequence data (not only DEGs) that includes both log2FC and p-value as a supplement table, which would be helpful for readers.

6. In Figure 3, although the authors explained briefly how they analyzed in-cohort sociability using Eco-HAB, a detailed explanation of the method in the Materials & Methods section may help to understand the analysis.

7. In Figure 3, not only Eco-HAB but also other familiar methods to analyze sociability tests (such as reciprocal or three-chamber social tests) should be included.

8. Not only male but also female homo mutant mice showed synaptic abnormalities, such as spine density (increased in males and decreased in females), as shown in Figure 4B and G, and synaptic transmission (increased in males and decreased in females) (Figure 4M and O). Regarding these results, the authors didn't explain why TRAP1 p.Q639 mutant caused opposite morphological and electrophysiological synaptic phenotypes between genders. The authors should clarify this point. In addition, according to these results, female homo mutant mice must have neurobehavioral abnormalities. However, no significant phenotypes were observed in females. The authors should explain and discuss this.

9. The relationship between Trap1 p.Q639 mutation and male-specific abnormalities is still unclear despite the ubiquitous expression of the Trap1 gene. Although Figure 5J and Figure S7 suggest male-specific mitochondrial abnormalities, the authors didn't explain or discuss why. The authors discuss this point.

10. In Figure 5J, the authors identified decreased use of succinate and increased consumption of pyruvate that produces NADH. In addition to assessing NADH and FADH₂ for electron flow, assessing acetyl Co-A activity is also essential.

11. The reviewer recommends the author analyze the complex I, II, and III activities that support the mitochondrial deficits (especially in the electron transport chain) of these mice.

Minor

1. Figure 2B: Please check the location of the PAM sequence. Not CTG, but TGG?

2. Figure 2D: Need to add molecular weight.

3. Although the scheme on the graph in Figure 5J is helpful for readers, the size is small. Please enlarge it and move it to Figure 5K or a supplementary figure.

Referee #3 (Remarks for Author):

In this manuscript, the authors identified a nonsense mutation in the TRAP1 gene in an ASD-discordant MZT. To investigate the underlying mechanism of this specific mutation in the TRAP1 gene which encodes a chaperone involved in mitochondrial protein homeostasis, they introduced the identical Trap1 mutation into the mouse genome using CRISPR-Cas9. Accordingly they found that newly generated knock-in Trap1 p.Q641 mice display ASD-related behavioral abnormalities exclusively in males. In addition, Trap1 mutation resulted in sex-specific changes in altered neuronal plasticity and dendritic spine morphology, number of presynaptic mitochondria, and metabolic substrate consumption although Trap1 mutation did not affect mitochondrial pre-RNA processing. Taken together, they argue that the TRAP1 mutation is an example of a monogenic ASD caused by impaired mitochondrial protein homeostasis. The manuscript was well written but I have some points to be considered to improve it.

Major points:

-In Fig 3 for in cohort sociability, sexual dimorphism they observed may come from non-Gaussian distribution of the male WT's instead of a behavioral defect of Trap1 mutants per se. They also need to carefully compare the behaviors between male and female WT. As an independent analysis of autistic like behavior, I would also suggest 3-chamber assay.

-In Discussion part, page 15, 2nd paragraph: they argue that sexual dimorphic phenomena can be explained by synaptic changes, but this is not true as there are many other possibilities as well and the causal link between synaptic changes and behaviors is not strong only based on their data. They need to tone down their argument.

-The authors need to discuss their hypothesis explaining why Trap1 mutation may cause sexual dimorphic phenotypes, which is a main theme of the manuscript.

Minor points:

-I'd suggest to move the NMDAR current data (Fig. S6 B,D) to the main figure, considering its importance.

-Citation errors? In Page 10, Fig 4L ->4M; 4M-> 4O; in page 11, Fig 4 N,O-> L,N

-Reference,

Many of human mutations in synaptic proteins have been functional characterized so far through whole genome sequencing, iPSC technology and animal model systems, etc. Compared to the vast majority of literatures in the field, this fact has not been properly introduced. I'd suggest to include some of the literatures in Introduction, for example,

Lim CS, et al. Dysfunction of NMDA receptors in neuronal models of an autism spectrum disorder patient with a DSCAM mutation and in Dscam-knockout mice. *Mol Psychiatry*. 2021;26:7538-7549.

Yoo J, et al., Shank mutant mice as an animal model of autism. *Philos Trans R Soc Lond B Biol Sci*. 2013; 369:20130143.

Point by point response to the Reviewer's comments

We thank the Reviewers for a positive opinion about our paper. We address all issues raised by Reviewers and edited the manuscript to improve its clarity. Below we provide a point by point response to the Reviewer's comments.

Referee #1 (Comments on Novelty/Model System for Author):
It is an interesting finding but no mechanism.

Referee #1 (Remarks for Author):

In this paper, Rydzanicz et al. identified a novel mutation of p.Q639 in the TRAP1 gene from patients with ASD. Since the TRAP1 gene encodes a mitochondria chaperone of the HSP90 family, these findings support mitochondrial deficit theory in ASD pathogenesis. Moreover, the mice with this homo mutation show not only synaptic and mitochondrial abnormalities but also ASD-like symptoms only in males. Sex difference in ASD is also one of the most critical issues not revealed. Thus, the manuscript would be of interest to many readers. However, at least in the current manuscript, mechanistic insights such as male-specific ASD-like symptoms and mitochondria deficits caused by TRAP1 gene mutation were lacking.

We thank the Reviewer for the appreciation of the potential broad interest in the results of our study. We also agree that mechanically, many questions remain unanswered.

However, with the addition of new experimental data and further controls, we believe our story - beginning with the identification of patients with a previously undescribed ASD-causing mutation and followed by the development of a new ASD animal model for which we provide functional data - is now better supported. Nevertheless, fully elucidating the pathomechanism of monogenic ASD associated with the TRAP1 mutation will require comprehensive and longitudinal studies, which extend beyond the scope of this initial paper.

1. In Figure 2D, the authors checked TRAP1 protein expression in mutant mice. However, it is unclear whether the point mutants p.Q639 generate truncated Trap1 protein. Therefore, the reviewer recommends the authors show a full-size WB image in a supplementary figure.
2. Related to #1, the authors should assess TRAP1 chaperone activity in the mutation.

As suggested by the Reviewer, we have provided a full-size anti-Trap1 western blot image, that shows that no signal for truncated Trap1 p.Q639 protein was detected. We also added a panel depicting the immunogenic region of Trap1 protein detected by the used antibody. New data is now presented in Appendix Figure S1.

Since the Trap1 p.Q639 mutation leads to complete downregulation of Trap1 protein expression we did not pursue the measurement of chaperone activity.

3. Sex-specific differences in cellular and behavioral deficits in the Trap1 p.Q639 mice are key findings of the study. Thus, assessing basal protein and mRNA expression levels of each genotype and sex is indispensable.

As suggested, we have added additional data on basal protein and mRNA expression levels of each genotype and sex - panels C and D on Figure 2 in the new, revised version of the manuscript.

4. In Figures 2F and G, the results of this expression analysis could not explain why TRAP1 point mutant causes male-specific ASD-like phenotypes as well as cellular-level deficits. The reviewer recommends the authors try a milder threshold. A proper threshold generates more DEGs, and following enrichment analysis may help to understand the core deficits of mutant mice from the viewpoint of gene expression.

We thank the Reviewer for pointing this out. We agree with the Reviewer that using a milder threshold in differential gene expression, especially in the exploratory phase of analysis, might expand the possibilities to uncover some molecular mechanisms underlying disease phenotypes. Thus, we have revisited our RNA-Seq data again.

First of all, to better visualize the change in *Trap1* gene expression and provide a summary of small changes in global transcriptome profiles, we decided to show the results as volcano plots (revised Figure 2 E, F and Figure EV2 A, B) instead of previously presented MA plots. Such visualization emphasize that we could detect only a few of differentially expressed genes when comparing WT with *Trap1*^{Q641*/Q641*} or *Trap1*^{WT/Q641*} and setting the threshold to $|\log_2(\text{FC})| > 1$, adj. P-value < 0.05. Additionally, we compared male to female samples of the same genotype grouped by *Trap1* mutation status (Figure EV2 C, D, E). There, as expected, the deregulated genes that showed up were the sex regulated loci (*Ddx3y*, *Uty*, *Eif2s3y* and *Xist*).

Moreover, following the Reviewer suggestion we explored the possibility of using milder thresholding conditions for selecting differentially expressed genes. Notably, quite uniform *P*-values distribution over the transcriptome without any peak close to 0 (Reviewer Figure 1) does not allow selection criteria to be chosen directly. Thus we analyzed the effect of several settings of *P*-value and fold change on the number of differentially expressed genes (Reviewer Table 1 below). With a list of DEGs when setting threshold from test 8. (*P*-value < 0.05, $|\log_2(\text{FC})| < 0.38$) GO enrichment analysis did not uncover any specific pathway deregulated under *Trap1* mutation. Moreover, slight upregulation of genes with mitochondrial localization under *Trap1* was observed (Figure EV2 F, G). Interestingly, the global view on transcriptional changes with mitochondrial localization showed that the profiles are very similar in both sexes except one small cluster differentiating (presented on Figure EV2 H).

Reviewer Figure 1. Exploratory analysis of RNA-Seq data. The distribution of P-values when testing of the difference between the mean of gene expression in *Trap1*^{WT} and *Trap1*^{Q641*/Q641*} is not equal 0.

Test No.	threshold	Total DEGs	UP-regulated	DOWN-regulated
1	adj. P - value < 0.05 & log ₂ (FC) > 1	3	2	1
2	log ₂ (FC) > 1	105	63	42
3	adj. P - value < 0.05	4	2	2
4	P - value < 0.00005	5	3	2
5	P - value < 0.0005	25	10	15
6	P - value < 0.05	882	368	514
7	P - value < 0.05 & log ₂ (FC) > 1	53	31	22
8	P - value < 0.05 & log ₂ (FC) > 0.38	341	177	164

Reviewer Table 1. Summary of the number of differentially expressed genes identified with different thresholds when comparing *Trap1*^{WT/Q641*} with *Trap1*^{WT} expressing male mice.

5. The authors should show the list of RNA-sequence data (not only DEGs) that includes both log₂FC and p-value as a supplement table, which would be helpful for readers.

We do agree that providing only the list of differentially expressed genes is not sufficient and that the table with the whole differential analysis should be supplied. We have added the data as Table EV1.

6. In Figure 3, although the authors explained briefly how they analyzed in-cohort sociability using Eco-HAB, a detailed explanation of the method in the Materials & Methods section may help to understand the analysis.

We thank the Reviewer for this remark. In the Materials & Methods section, we have added a detailed explanation of in-cohort sociability.

7. In Figure 3, not only Eco-HAB but also other familiar methods to analyze sociability tests (such as reciprocal or three-chamber social tests) should be included.

As suggested by the Reviewer, we have additionally validated our behavioral data obtained using Eco-HAB with a standard three-chamber social approach task, which tests the preference for an unfamiliar mouse over an inanimate object. The test revealed that both mutant males and females show decreased interest in an unfamiliar animal. In contrast, the Eco-HAB data presented in the original manuscript show a social deficit only in males. This test assesses how much time mice spend voluntarily with familiar animals, living together in one cage. Thus, we conclude that the behavioral phenotype is more pronounced in males than in females, as males exhibit problems with social interaction with both unfamiliar and familiar animals, in contrast to females, who show deficits only when interacting with unfamiliar conspecifics. This observation resembles the experience of many autistic individuals, who often find interacting with unfamiliar people more stressful than interacting with familiar ones. However, some autistic people might still find interactions with familiar individuals challenging. The new data are now presented in Figure 3 and Figure EV3 and discussed in the manuscript.

8. Not only male but also female homo mutant mice showed synaptic abnormalities, such as spine density (increased in males and decreased in females), as shown in Figure 4B and G, and synaptic transmission (increased in males and decreased in females) (Figure 4M and O). Regarding these results, the authors didn't explain why TRAP1 p.Q639 mutant caused opposite morphological and electrophysiological synaptic phenotypes between genders. The authors should clarify this point. In addition, according to these results, female homo mutant mice must have neurobehavioral abnormalities. However, no significant phenotypes were observed in females. The authors should explain and discuss this.

We agree with this remark, especially in the light of the new social approach experiments showing deficits on both males and females. We have made substantial changes in the manuscript starting from the change of the title. We also extended the discussion highlighting the possible hypothesis that can explain the more pronounced neurobehavioral phenotypes observed in males.

9. The relationship between Trap1 p.Q639 mutation and male-specific abnormalities is still unclear despite the ubiquitous expression of the Trap1 gene. Although Figure 5J and Figure S7 suggest male-specific mitochondrial abnormalities, the authors didn't explain or discuss why. The authors discuss this point.

To answer Reviewers' concern we extended the Discussion. We realize that understanding the exact cause of the male bias in the *Trap1*^{Q641*/Q641*} mice will require further investigation into the mitochondrial functions in neurons and with the specific focus on differences between genders.

10. In Figure 5J, the authors identified decreased use of succinate and increased consumption of pyruvate that produces NADH. In addition to assessing NADH and FADH2 for electron flow, assessing acetyl Co-A activity is also essential.

We agree with the Reviewer, that in addition to assessing mitochondrial function by measuring the rates of electron flow using MitoPlatesTM (Biolog), other additional experiments to support the observed mitochondrial deficits in *Trap1*^{Q641*/Q641*} mice would be beneficial. As described below, we decided to use High resolution respirometry (Oroboros) to strengthen our conclusions.

We agree that analyzing levels of certain metabolites would give us additional insight into observed mitochondrial deficits. Indeed, in the future we plan to perform metabolomic analysis, however these experiments are beyond the study presented in the current manuscript.

11. The reviewer recommends the author analyze the complex I, II, and III activities that support the mitochondrial deficits (especially in the electron transport chain) of these mice.

We thank the Reviewer for this comment. We performed additional experiments to analyze the respiration of brain mitochondria more precisely and measure the activity of respirometry chain complexes. We used high-resolution respirometry technique that allows measuring O₂ consumption rate in mitochondria isolated from the brain of WT and *Trap1*^{Q641*/Q641*} mice.

We found that in the presence of pyruvate and malate, which provide electrons to complex I, the mitochondria from *Trap1*^{Q641*/Q641*} mice respired at a significantly higher rate than the wild types. This was true for males and females. Moreover, respiration of mitochondria from male *Trap1*^{Q641*/Q641*} mice was slightly decreased (tendency) in the presence of ascorbate and TMPD that provide electrons directly to complex IV. New data was added to the manuscript and is presented on new panels (J, K, L) on Figure 5.

Minor

1. Figure 2B: Please check the location of the PAM sequence. Not CTG, but TGG?

We apologize for this editorial error, location of the PAM sequence was corrected in the revised version of the Figure 2B.

2. Figure 2D: Need to add molecular weight.

The Figure was corrected accordingly.

3. Although the scheme on the graph in Figure 5J is helpful for readers, the size is small. Please enlarge it and move it to Figure 5K or a supplementary figure.

Figure 5 was significantly changed. The scheme was enlarged as suggested and is now a part of Figure EV5.

Referee #3 (Remarks for Author):

In this manuscript, the authors identified a nonsense mutation in the TRAP1 gene in an ASD-discordant MZT. To investigate the underlying mechanism of this specific mutation in the TRAP1 gene which encodes a chaperone involved in mitochondrial protein homeostasis, they introduced the identical *Trap1* mutation into the mouse genome using CRISPR-Cas9. Accordingly they found that newly generated knock-in *Trap1* p.Q641 mice display ASD-related behavioral abnormalities exclusively in males. In addition, *Trap1* mutation resulted in sex-specific changes in altered neuronal plasticity and dendritic spine morphology, number of presynaptic mitochondria, and metabolic substrate consumption although *Trap1* mutation did not affect mitochondrial pre-RNA processing. Taken together, they argue that the TRAP1 mutation is an example of a monogenic ASD caused by impaired mitochondrial protein

homeostasis. The manuscript was well written but I have some points to be considered to improve it.

We appreciate the overall positive opinion about our manuscript.

Major points:

-In Fig 3 for in cohort sociability, sexual dimorphism they observed may come from non-Gaussian distribution of the male WT's instead of a behavioral defect of Trap1 mutants per se. They also need to carefully compare the behaviors between male and female WT. As an independent analysis of autistic like behavior, I would also suggest 3-chamber assay.

As the distributions of in-cohort sociability usually do not follow a Gaussian distribution, we apply non-parametric statistics, specifically the Kolmogorov–Smirnov test, which makes minimal assumptions about the underlying distribution. Thus, we believe that the conclusions of this test are accurate. However, as suggested by the Reviewer, we decided to further investigate the social deficits of the Trap1 mice using the three-chamber test. We used a standard three-chamber social approach task, which tests the preference for an unfamiliar mouse over an inanimate object. The test revealed that both mutant males and females show decreased interest in an unfamiliar animal. In contrast, the Eco-HAB data presented in the original manuscript show a social deficit only in males. This test assesses how much time mice spend voluntarily with familiar animals, living together in one cage. Thus, we conclude that the behavioral phenotype is more pronounced in males than in females, as males exhibit problems with social interaction with both unfamiliar and familiar animals, in contrast to females, who show deficits only when interacting with unfamiliar conspecifics. This observation resembles the experience of many autistic individuals, who often find interacting with unfamiliar people more stressful than interacting with familiar ones. However, some autistic people might still find interactions with familiar individuals challenging. The new data are now presented in Figure 3 and Figure EV3 and discussed in the manuscript.

-In Discussion part, page 15, 2nd paragraph: they argue that sexual dimorphic phenomena can be explained by synaptic changes, but this is not true as there are many other possibilities as well and the causal link between synaptic changes and behaviors is not strong only based on their data. They need to tone down their argument.

We agree with this remark, especially in the light of the new social approach experiments showing deficits on both males and females. We have made substantial changes in the manuscript starting from the change of the title.

-The authors need to discuss their hypothesis explaining why Trap1 mutation may cause sexual dimorphic phenotypes, which is a main theme of the manuscript.

Discussion was extended and the hypothesis explaining sex differences in *Trap1*^{Q641*/Q641*} mice behavioral and molecular phenotypes were discussed on page 12.

Minor points:

-I'd suggest to move the NMDAR current data (Fig. S6 B,D) to the main figure, considering its importance.

-Citation errors? In Page 10, Fig 4L ->4M; 4M-> 4O; in page 11, Fig 4 N,O-> L,N

We thank the Reviewer for this suggestion, panels B, D from the supplementary Fig. S6 were moved to the main Figure 4. Also, the order of panels K-P was adjusted and the citation errors were corrected.

-Reference,

Many of human mutations in synaptic proteins have been functional characterized so far through whole genome sequencing, iPSC technology and animal model systems, etc. Compared to the vast majority of literature in the field, this fact has not been properly introduced. I'd suggest to include some of the literatures in Introduction, for example, Lim CS, et al. Dysfunction of NMDA receptors in neuronal models of an autism spectrum disorder patient with a DSCAM mutation and in Dscam-knockout mice. *Mol Psychiatry*. 2021;26:7538-7549.

Yoo J, et al., Shank mutant mice as an animal model of autism. *Philos Trans R Soc Lond B Biol Sci*. 2013; 369:20130143.

Thank you for bringing to our attention these very interesting and relevant papers that we cited in the Introduction.

We appreciate your careful evaluation of our work that helped us to improve the quality of the paper. We hope that this revision meets with your approval. We have included the revised manuscript version that highlights the changes from the original submission (in blue).

Thanks again for your interest in our work. We await your review of our revised manuscript.

2nd Sep 2024

Dear Dr. Dziembowska,

Thank you for the submission of your revised manuscript to EMBO Molecular Medicine. I am pleased to inform you that we will be able to accept your manuscript pending the following final amendments:

- 1) We note that you currently have together with you, a total of 3 co-corresponding authors. Is that correct? Do you confirm equal contribution of these authors, able to take full responsibility for the paper and its content? While there is no limit per se to the number of co-corresponding authors, 3 co-corresponding authors is rather rare, and may not reflect as intended to the community.
- 2) In the main manuscript file, please do the following:
 - Please address all comments suggested by our data editors listed below:
 - o Figure legends:
 1. Please note that the legends for figures 4b-g; l-p is not provided in the sequential manner (legend for figures 4f, g is provided before legend of figures 4b-e, 4c-e; legend for figures 4n, o is provided before legend of figures 4l-m, 4m, p). This needs to be rectified.
 2. Please note that the figure EV 1c is missing in the manuscript. This needs to be rectified.
 3. Please note that the exact p values are not provided in the legends of figures 2d; 3d-e, g-h; 4b-e, g-o; 5g, k-l; EV 3d; EV 5c.
 4. Please indicate the statistical test used for data analysis in the legends of figures 2e-f; EV 2a-h.
 5. Please note that in figures 2c; EV 3g; EV 5c; there is a mismatch between the annotated p values in the figure legend and the annotated p values in the figure file that should be corrected.
 6. Please note that information related to n is missing in the legends of figures 5h-i.
 7. Please note that the error bars are not defined in the legends of figures 3d-e, g-h; 4k-p; EV 3c-d; EV 4b-c.
 8. Please note that the scale bar needs to be defined for figures 5d-f.
 - Add up to 5 keywords.
 - Add callouts for Fig EV1 and EV3. There are callouts for Fig S1 and Fig S4A, please correct. Also, where appropriate please add panels to the callouts.
 - In Methods, add the following paragraph:

Graphics:

(some of the... OR Figure #... OR synopsis) Graphics were created with BioRender.com.

Please remove the reference to BioRender from Acknowledgments.

- In Methods, add a statistical paragraph that should reflect all information that you have filled in the Authors Checklist, especially regarding randomization, blinding, replication.
- Indicate in legends exact n and exact p values, not a range, along with the statistical test used. To keep the figures "clear" some authors found providing an Appendix table Sx with all exact p-values preferable. You are welcome to do this if you want to.
- Please include structured Methods section that includes a Reagents and Tools Table (should be uploaded as a separate file) followed by a Methods and Protocols section. More information on how to adhere to this format as well as downloadable templates (.docx) for the Reagents and Tools Table can be found in our author guidelines:
<https://www.embopress.org/page/journal/17574684/authorguide#structuredmethods>
- An example of a paper with Structured Methods can be found here:
<https://www.embopress.org/doi/full/10.1038/s44320-024-00037-6#sec-4>
- In Methods, provide the statement that informed consent was obtained from all human subjects and confirm that the experiments conformed to the principles set out in the WMA Declaration of Helsinki and the Department of Health and Human Services Belmont Report.
- Rename "Competing interests" to "Disclosure Statement & Competing Interests". We updated our journal's competing interests policy in January 2022 and request authors to consider both actual and perceived competing interests. Please review the policy <https://www.embopress.org/competing-interests> and update your competing interests if necessary.
- Correct the reference citation in the text and reference list. In the text, a reference should be cited by author and year of publication. Include a space between a word and the opening parenthesis of the reference that follows. In the reference list, citations should be listed in alphabetical order. Where there are more than 10 authors on a paper, 10 will be listed, followed by "et al.". Please also move table references to the main reference list. Please check "Author Guidelines" for more information.
<https://www.embopress.org/page/journal/17574684/authorguide#referencesformat>
- Rename "Data and materials availability" to "Data availability" and remove all information related to statistics that should be placed in the statistical paragraph mentioned above. Leave only explanation about the ethical restriction and deposited RNA Seq data. Please use the following format to report the accession number of your data:

[data type]: [full name of the resource] [accession number/identifier] [(doi or URL or identifiers.org/DATABASE:ACCESSION)]

-

Please check "Author Guidelines" for more information.

<https://www.embopress.org/page/journal/17574684/authorguide#availabilityofpublishedmaterial>

3) Tables: Rename Table EV1 to Dataset EV1 and add its legend to the file in a separate tab/worksheet. Please update the callout in the main text. Place main tables with their legends after Figure legends in the main manuscript file.

4) Funding: Please merge it with "Acknowledgments".

5) The Paper Explained: Please provide "The Paper Explained" and add it to the main manuscript text. Please check "Author Guidelines" for more information. <https://www.embopress.org/page/journal/17574684/authorguide#researcharticleguide>

6) Synopsis: Every published paper now includes a 'Synopsis' to further enhance discoverability. Synopses are displayed on the journal webpage and are freely accessible to all readers. They include separate synopsis image and synopsis text.

- Synopsis image: Please reformat the image to 550 px-wide x (250-400)-px high and upload it as a high-resolution jpeg file.

- Synopsis text: Please provide a short standfirst (maximum of 300 characters, including space) as well as 2-5 one sentence bullet points that summarise the paper as a .doc file. Please write the bullet points to summarise the key NEW findings. They should be designed to be complementary to the abstract - i.e. not repeat the same text. We encourage inclusion of key acronyms and quantitative information (maximum of 30 words / bullet point). Please use the passive voice.

7) As part of the EMBO Publications transparent editorial process initiative (see our Editorial at

<http://embomolmed.embopress.org/content/2/9/329>), EMBO Molecular Medicine will publish online a Review Process File (RPF) to accompany accepted manuscripts. This file will be published in conjunction with your paper and will include the anonymous referee reports, your point-by-point response and all pertinent correspondence relating to the manuscript. Let us know whether you agree with the publication of the RPF and as here, if you want to remove or not any figures from it prior to publication.

8) Please provide a point-by-point letter INCLUDING my comments as well as the reviewer's reports and your detailed responses (as Word file).

I look forward to reading a new revised version of your manuscript as soon as possible.

Yours sincerely,

Zeljko Durdevic

*** Instructions to submit your revised manuscript ***

1) a .docx formatted version of the manuscript text (including Figure legends and tables)

2) Separate figure files*

3) supplemental information as Expanded View and/or Appendix. Please carefully check the authors guidelines for formatting Expanded view and Appendix figures and tables at <https://www.embopress.org/page/journal/17574684/authorguide#expandedview>

4) a letter INCLUDING the reviewer's reports and your detailed responses to their comments (as Word file).

5) The paper explained: EMBO Molecular Medicine articles are accompanied by a summary of the articles to emphasize the major findings in the paper and their medical implications for the non-specialist reader. Please provide a draft summary of your article highlighting

This may be edited to ensure that readers understand the significance and context of the research.

Please refer to any of our published articles for an example.

6) For more information: There is space at the end of each article to list relevant web links for further consultation by our readers. Could you identify some relevant ones and provide such information as well? Some examples are patient associations, relevant databases, OMIM/proteins/genes links, author's websites, etc...

7) Author contributions: the contribution of every author must be detailed in a separate section.

8) EMBO Molecular Medicine now requires a complete author checklist (<https://www.embopress.org/page/journal/17574684/authorguide>) to be submitted with all revised manuscripts. Please use the checklist as guideline for the sort of information we need WITHIN the manuscript. The checklist should only be filled with page numbers where the information can be found. This is particularly important for animal reporting, antibody dilutions (missing) and exact values and n that should be indicated instead of a range.

9) Every published paper now includes a 'Synopsis' to further enhance discoverability. Synopses are displayed on the journal webpage and are freely accessible to all readers. They include a short stand first (maximum of 300 characters, including space) as well as 2-5 one sentence bullet points that summarise the paper. Please write the bullet points to summarise the key NEW findings. They should be designed to be complementary to the abstract - i.e. not repeat the same text. We encourage inclusion of key acronyms and quantitative information (maximum of 30 words / bullet point). Please use the passive voice. Please attach these in a separate file or send them by email, we will incorporate them accordingly.

You are also welcome to suggest a striking image or visual abstract to illustrate your article. If you do please provide a jpeg file 550 px-wide x 300-600px high.

10) A Conflict of Interest statement should be provided in the main text

11) Please note that we now mandate that all corresponding authors list an ORCID digital identifier. This takes <90 seconds to complete. We encourage all authors to supply an ORCID identifier, which will be linked to their name for unambiguous name identification.

Currently, our records indicate that the ORCID for your account is 0000-0002-0650-6438.

Link Not Available

12) Include a Reagents and Tools Table as part of the Methods section, which can be downloaded from our author guidelines (<https://www.embopress.org/page/journal/17574684/authorguide#structuredmethods>)

Photos 400-800 DPI

*Additional important information regarding figures and illustrations can be found at

<https://bit.ly/EMBOPressFigurePreparationGuideline>. See also figure legend preparation guidelines:

<https://www.embopress.org/page/journal/17574684/authorguide#figureformat>

***** Reviewer's comments *****

Referee #1 (Comments on Novelty/Model System for Author):

N/A

Referee #1 (Remarks for Author):

All additional experiments the reviewer commented on have been performed.

Referee #3 (Remarks for Author):

The authors addressed my concerns.

The authors addressed the minor editorial issues.

16th Sep 2024

Dear Dr. Dziembowska,

We are pleased to inform you that your manuscript is accepted for publication and is now being sent to our publisher to be included in the next available issue of EMBO Molecular Medicine.
